# ALPAGASUS: TRAINING A BETTER ALPACA WITH FEWER DATA

**Lichang Chen**[*†], **Shiyang Li** [*‡], **Jun Yan**[♯], **Hai Wang** [‡], **Kalpa Gunaratna**[‡], **Vikas Yadav**[‡],
**Zheng Tang**[‡], **Vijay Srinivasan**[‡], **Tianyi Zhou**[†], **Heng Huang**[†], **Hongxia Jin**[‡]
[†] University of Maryland, College Park [‡] Samsung Research America [♯] University of Southern California
{bobchen, tianyi, heng}@umd.edu
{shiyang.li, h.wang2, k.gunaratna, vikas.y, zheng.tang,
v.srinivasan, hongxia.jin}@samsung.com
yanjun@usc.edu

## ABSTRACT

Large language models (LLMs) strengthen instruction-following capability through instruction-finetuning (IFT) on supervised instruction/response data. However, widely used IFT datasets (e.g., ALPACA's 52k data) surprisingly contain many low-quality instances with incorrect or irrelevant responses, which are misleading and detrimental to IFT. In this paper, we propose a simple and effective data selection strategy that automatically identifies and filters out low-quality data using a strong LLM (e.g., ChatGPT). To this end, we introduce ALPAGASUS, which is finetuned on only 9k high-quality data filtered from the 52k ALPACA data. ALPAGASUS significantly outperforms the original ALPACA as evaluated by GPT-4 on multiple test sets and the controlled human evaluation. Its 13B variant matches $> 90\%$ performance of its teacher LLM (i.e., Text-Davinci-003 generating the 52k data) on test tasks. It also provides 5.7x faster training, reducing the training time for a 7B variant from 80 minutes (for ALPACA) to 14 minutes [1]. Moreover, the experiments prove the efficacy of our method across diverse datasets, base models, and LLM filters. Overall, ALPAGASUS demonstrates a novel data-centric IFT paradigm that can be generally applied to instruction-tuning data, leading to faster training and better instruction-following models. Our project page is available at:
https://lichang-chen.github.io/AlpaGasus/.

## 1 INTRODUCTION

Instruction fine-tuning (IFT) (Longpre et al., 2023) has been recently applied as an essential continual training stage for pre-trained large language models (LLMs) to achieve instruction-following capability (Ouyang et al., 2022b; Chen et al., 2023b), which is often attributed to aligning the models' behavior with a diverse set of human instructions and responses (Taori et al., 2023; Askell et al., 2021). The recent series of open-sourced instruction-tuned models (Taori et al., 2023; Xu et al., 2023) reveal that the alignment of better IFT data could result in better instruction-following skills. For example, GPT-4-LLM (Peng et al., 2023) (with GPT-4 (OpenAI, 2023b) as its teacher) exhibits better reasoning and math ability than ALPACA (Taori et al., 2023) (with Text-davinci-003 as its teacher), though they share the same base model LLaMA (Touvron et al., 2023), demonstrating the importance of data quality.

Although stronger teachers can usually bring further improvement by providing better IFT data, their responses inevitably include incorrect or irrelevant answers to the corresponding instructions (see examples in Fig. 2), which can be misleading or detrimental to IFT. Moreover, these data also increase unnecessary training costs. Alpaca-cleaned[2] is the pioneer of filtering bad data in ALPACA dataset though it requires humans fully involved in examining and filtering the data. Nonetheless, how to automatically filter out poor-quality data from IFT datasets has not been investigated yet. A primary

---

[*]Equal Contribution. This work was done when Lichang Chen and Jun Yan interned at Samsung Research America.

[1]We apply IFT for the same number of epochs as ALPACA(7B) but on fewer data, using 4×NVIDIA A100 (80GB) GPUs and following the original ALPACA setting and hyperparameters.

[2]https://github.com/gururise/AlpacaDataCleaned/

bottleneck is that rating the data quality usually requires expensive human labor but still may not be accurate for IFT because stronger teachers are more powerful in generating eloquent but incorrect responses that are more subtle to detect by humans. When considering datasets crafted by humans, such as the Dolly dataset (Dolly, 2023), assessing quality becomes even more intricate, given that responses stem from seasoned writers.

This paper aims to bridge the gap by proposing a novel data-filtering strategy for IFT that is efficient, automatic, and accurate. Specifically, we design a prompt applied to a powerful LLM (e.g., ChatGPT) for evaluating the quality of each (instruction, input, response) tuple and then filter out the ones with scores lower than a threshold. By applying this filter to the 52k data used to train ALPACA, we find that a majority of the data suffer from low-quality issues. Using the LLM filter, IFT on a much smaller but carefully filtered subset of 9k data produces a much better model, i.e., ALPAGASUS, than the original ALPACA, as shown in Fig. 1, following exactly the same training configuration of ALPACA. This also reduces the training time from 80 minutes to merely 14 minutes on $4\times$ NVIDIA A100 (80GB) GPUs. Moreover, we validate the versatility of our method, demonstrating its effectiveness on a range of datasets(e.g., Dolly, Alpaca, GPT4LLM), base models(e.g., LLaMA-1 and LLaMA-2), and LLM filters(e.g., ChatGPT and Claude-2). This discovery is inspiring, as it shows that the data quality in IFT can outweigh the quantity. In addition, this shift towards prioritizing data quality presents a new and more efficient paradigm that can generally improve the fine-tuning of LLMs.

Our experiments include comprehensive evaluations for our ALPAGASUS, incorporating free-form instruction evaluation, various benchmarks, and human studies. We select four different human-instruction test sets for evaluating instruction-following capability, including the ones used by WizardLM (Xu et al., 2023), Vicuna (Chiang et al., 2023), Koala (Geng et al., 2023), and Self-Instruct (Wang et al., 2022). Given the notable advantages that GPT-4 judge could match with both the controlled and crowdsourced human preferences ($> 80\%$ agreement) (Zheng et al., 2023), we employ GPT-4 as our judge for the major evaluations. In the 7B and 13B model comparisons, ALPAGASUS performs significantly better than ALPACA on all four test sets. To address potential concerns regarding biases in model-based evaluations, we conduct human studies and benchmark evaluations, both of which corroborate the superiority of our model compared to baseline counterparts. Furthermore, we present a fine-grained evaluation of ALPAGASUS on individual tasks including Generic, Roleplay, Knowledge, and Commonsense from the Vicuna test set. The results indicate ALPAGASUS exhibits advantages on a majority of the tasks.

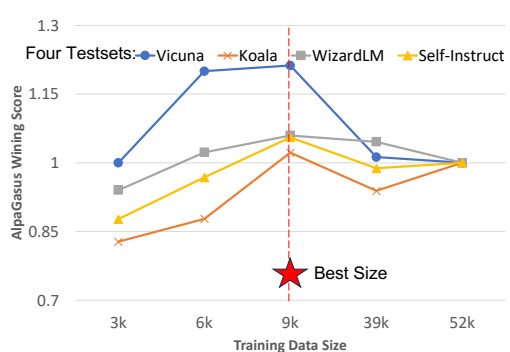

Figure 1: Performance of ALPAGASUS on four test sets when increasing its finetuning data, where the winning score is $\frac{\#\text{Win}-\#\text{Lose}}{\#\text{Testset}} + 1$ with #Testset = #Win + #Tie + #Lose to be the test set size and #Win/#Tie/#Lose to be the number of samples on which ALPAGASUS wins/ties/loses compared to ALPACA 52K.

To sum up, our data-filtering approach exhibits significant benefits in terms of scalability and automation. We also demonstrate that prudent management of training data quality can lead to substantial performance improvement and computation savings of IFT. In addition, our data selection and evaluation strategies can generalize to other instruction finetuning datasets and LLMs, thereby paving the way for a promising new research trajectory aimed at pragmatic LLM deployment.

## 2 METHODOLOGY

### 2.1 OVERVIEW

Unlike the recent work (Zhou et al., 2023), which relies on human labor to curate 1k high-quality instruction data that leads to a better finetuned model, we aim to avoid the expensive and time-consuming human annotations. Hence, we exploit the potential of strong LLMs to be auto-graders of the training data and then filter out the data with lower scores.

In particular, we prompt a strong API LLM, i.e., ChatGPT, to produce a score for each triplet of (instruction, input, response). The prompt is given in Fig. 3, where "dimension" denotes a

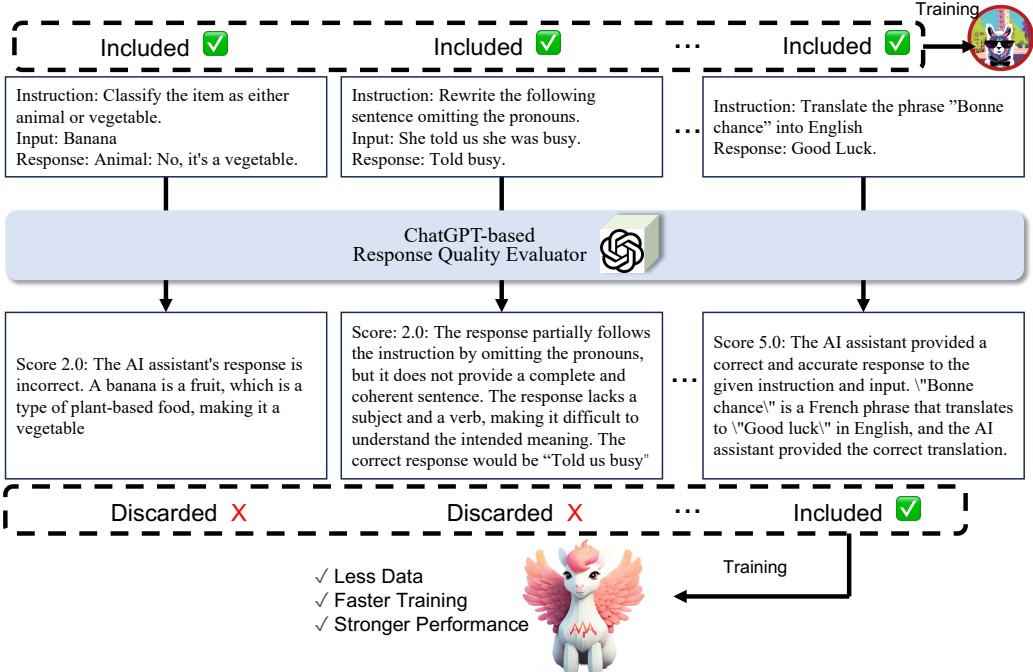

Figure 2: The fine-tuning pipeline of ALPAGASUS. We prompt ChatGPT as our auto-grader to score each training triplet on a scale of 0 to 5. We then use the exact same instruction fine-tuning script of ALPACA to train ALPAGASUS on the filtered data with scores higher than a threshold.

---

**System Prompt:**
We would like to request your feedback on the performance of AI assistant in response to the instruction and the given input displayed following.

Instruction: [Instruction]
Input: [Input]
Response: [Response]

**User Prompt:**
Please rate according to the [dimension] of the response to the instruction and the input. Each assistant receives a score on a scale of 0 to 5, where a higher score indicates higher level of the [dimension]. Please first output a single line containing the value indicating the scores. In the subsequent line, please provide a comprehensive explanation of your evaluation, avoiding any potential bias.

---

Figure 3: Prompt $p_G$ to ChatGPT for rating and filtering training data in Eq. (1).

user-preferred property such as helpfulness and accuracy. We then only select the triplets with scores higher than a certain threshold to fine-tune a LLaMA-series model following an existing IFT pipeline. Fig. 2 illustrates the data selection and training pipeline.

## 2.2 DATA RATING AND FILTERING

Given an IFT dataset $V$ of triplets $x =$(instruction, input, response) with $x \in V$ and an open-sourced LLM $\theta$ (e.g., LLaMA), let $\theta_V$ denote the finetuned $\theta$ on $V$, our overarching goal is to select a subset $S \subset V$ such that IFT on $S$ results in a better model $\theta_S$ than $\theta_V$.

In order to select $S$ from $V$, we prompt an API LLM $G(\cdot)$ (e.g., ChatGPT[3]) as an auto-grader rating each sample $x \in V$ by a score $G(x, p_G)$ wherein $p_G$ is the rating prompt in Fig. 3. We then select $x_i$ whose score is above a certain threshold $\tau$, i.e.,

$$S \triangleq \{x \in V : G(x, p_G) \geq \tau\}. \tag{1}$$

We achieve $\theta_S$ by finetuning $\theta$ on $S$ using an existing IFT framework.

---

[3]We also use claude-2 as our response quality evaluator, which can be found in Appendix A.2

## 2.3 ALPAGASUS: 9K TRAINING DATA FILTERED FROM ALPACA

For "dimension" in the rating prompt $p_G$ shown in Fig. 3, given that "accuracy" closely aligns with human expectations of LLMs' responses, we designate "accuracy" as the dimension for rating purposes.[4] Correspondingly, we establish $\tau$ in Eq. (1) as an accuracy threshold for the subsequent experiments. The distribution of scores in relation to the 52k Alpaca dataset is presented in Fig. 4.

In particular, we choose the threshold $\tau = 4.5$ according to the score histogram. For the ALPACA dataset $V$ with 52,002 samples, this filtering criterion leads to a subset $S$ of 9,229 samples [5].

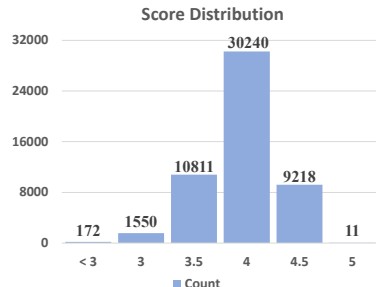

Figure 4: Histogram of Scores (Alpaca Dataset).

## 3 EXPERIMENTAL SETUP

### 3.1 FREE-FORM INSTRUCTION EVALUATION

Most instruction-tuned models are evaluated on one test set that might not cover sufficient diverse instructions and thus leads to a risk of biased evaluation (Chia et al., 2023). To conduct a holistic evaluation of ALPAGASUS, we curate our test sets from Self-instruct (Wang et al., 2022), Vicuna (Chiang et al., 2023), WizardLM (Xu et al., 2023), and Koala (Geng et al., 2023), which together can cover more types of instructions and reduce the evaluation bias. Details of these four test sets are provided in Table 1.

### 3.2 BASELINE MODELS

We compare our ALPAGASUS with the following four recent LLMs.

ALPACA    (Taori et al., 2023) is an open-sourced model developed by Stanford University through IFT of LLaMA on a training dataset of 52,002 (instruction, input, response) samples with the responses generated by Text-Davinci-003 (teacher).

| Test Set | # Samples | Category |
|----------|-----------|----------|
| Koala | 180 | |
| Vicuna | 80 | ✓ |
| WizardLM | 218 | ✓ |
| Self-Instruct | 252 | |

Table 1: Four test sets used in this paper.

TEXT-DAVINCI-003    is an OpenAI LLM trained with an increased emphasis on contextual understanding and response accuracy. Its proficiency in capturing complex linguistic patterns makes it a powerful teacher LLM for generating high-quality training data for finetuning LLMs such as ALPACA.

CHATGPT    (OpenAI, 2023a) is an AI chatbot finetuned via reinforcement learning with human feedback (RLHF). It exhibits exceptional capability across a wide range of tasks and might be the most popular chatbot recently. Hence, it would be interesting to study to what extent ALPAGASUS can match its performance.

CLAUDE    (Bai et al., 2022) is an AI chatbot developed by Anthropic. It was finetuned by RLHF to align with humans' preference on three dimensions, i.e., helpful, honest, and harmless. We use Claude-v1.1 for comparison, which is comparable to ChatGPT on the AlpacaEval (Li et al., 2023).

### 3.3 EVALUATION METRICS

The evaluation of the instruction-following capability of LLMs is usually challenging due to the existence of multiple eligible responses to one instruction and the difficulty of reproducing human evaluations. In light of the recent advancements in automated evaluation (Dubois et al., 2023; Zheng et al., 2023; Chiang et al., 2023), which offer superior scalability and explainability than human studies, we also apply an API LLM $J(\cdot)$ (e.g., GPT-4) as the judge to evaluate $\theta_S$ and compare it with $\theta_V$. In particular, we apply $J(\cdot)$ to compare the responses of $\theta_S$ and $\theta_V$ to each instruction $z$ drawn from a test set $D$. Let $F(z; \theta_V)$ and $F(z; \theta_S)$ denote the two models' responses to instruction $z \in D$,

---

[4]We defer the experiment of other dimensions, e.g., helpfulness, to the Appendix A.5.
[5]52k denotes 52002 samples from the original Alpaca training set and 9k represents 9229 data samples. (either randomly sampled or filtered in our experiments)

the judge outputs a score for each response and we aim to achieve a higher score on $\theta_S$, i.e.,

$$J(F(z; \theta_S)) \geq J(F(z; \theta_V)) \tag{2}$$

for most $z \in D$. In our experiments, we include both models' responses in the input to the judge (e.g., GPT-4), followed by an instruction to the judge, which aims to rate the responses with a score between 1 and 10. Details of the input and prompt to the judge can be found in Appendix C[6]

Since there exists position bias within LLM judges, which refers to a phenomenon where LLM judges have tendencies to prefer specific positions over others (Wang et al., 2018; Ko et al., 2020; Wang et al., 2023), to mitigate it, we try both orders (i.e., placing ALPAGASUS's response before/after the baseline model's response) and define the final judge of "Win-Tie-Lose" to be:(1) **Win:** ALPAGASUS wins twice, or wins once and draws once. (2) **Tie:** ALPAGASUS draws twice, or wins once and loses once. (3) **Lose:** ALPAGASUS loses twice, or loses once and draws once. To avoid cut-off responses, we allow models to generate up to 1024 tokens. For ChatGPT, Claude, and Text-Davinci-003, we set the temperature to 0.0, respectively, to reduce randomness and ensure a fair comparison.

## 4 EXPERIMENTAL RESULTS

### 4.1 QUALITY MATTERS MORE THAN QUANTITY

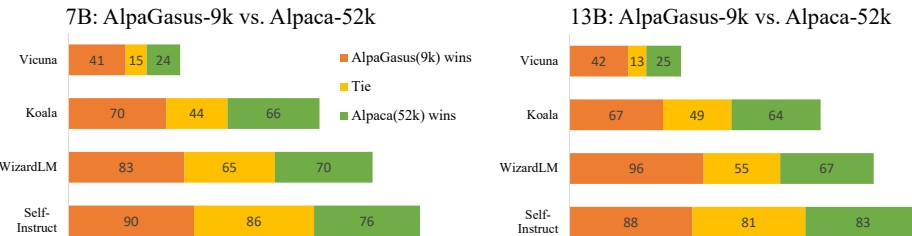

Figure 5: **Main results:** comparing ALPAGASUS and ALPACA on their 7B and 13B models. ALPAGASUS-9k achieves much better performance than ALPACA-52k on all four test sets: Vicuna, Koala, Self-Instruct, and WizardLM.

**AlpaGasus-9k vs. Alpaca-52k** We compare ALPAGASUS and ALPACA on two sizes of models in Fig. 5. They only differ in the training data: ALPACA uses all the 52k data while ALPAGASUS only uses 9k data selected from the 52k. Their hyperparameters and training scripts are the same. As shown in the evaluation results, ALPAGASUS significantly outperforms the original ALPACA across all four test sets. Moreover, when using LLaMA-2 as the base model, we observe consistent outcomes (See Appendix A.3). This consistency underscores the universality of our data filtering method, irrespective of the model choices. These findings also confirm that our training data selection approach leads to superior performance even when the selected training data are only 17.75% of the original dataset.

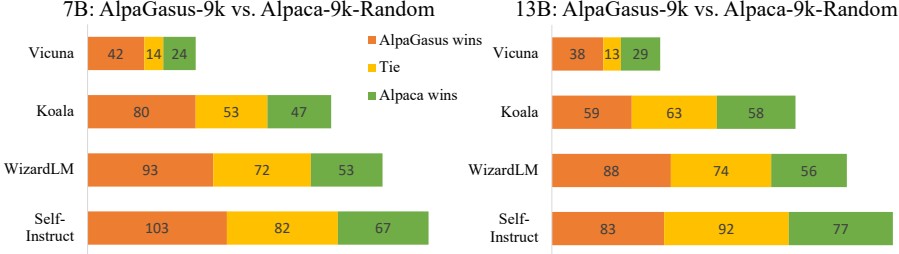

Figure 6: Comparing ALPAGASUS with LLaMA finetuned on **randomly selected data**.

---

[6]To address potential concerns regarding bias in the evaluation prompts, we also present results of using alternative evaluation prompts in Appendix A.1.

**Quality-Guided Filtering vs. Random Filtering** To investigate the efficacy of our data selection strategy, we compare ALPAGASUS with LLaMA models fine-tuned on a randomly sampled subset of the ALPACA 52k data, denoted by ALPACA-9k-random in Fig. 6. Both models start from the same initial model (i.e., LLaMA) and are then finetuned on the same number of samples (i.e., 9k). They only differ in terms of the data selection criteria. In Fig. 6, we compare the two types of models under two model sizes, i.e., 7B and 13B. ALPAGASUS-9k significantly outperforms ALPACA-9k-random, showing the high quality of our selected data and their importance to the performance of IFT.

## 4.2 HOW MUCH DATA SHOULD BE FILTERED?

**Threshold $\tau$ of data filtering.** In Eq. (1), we select data with score$\geq \tau$ and we set $\tau = 4.5$ in our main experiments, which results in 9k out of the 52k data to finetune ALPAGASUS. To study the impact of the threshold $\tau$ on IFT, we compare ALPAGASUS with LLaMA finetuned on 39k data selected by applying a lower threshold of $\tau = 4.0$. We report the comparison results in Fig. 7. When tested on the Koala and WizardLM test sets, ALPACA-39k model outperforms the original ALPACA-52k model. However, when using the Vicuna and Self-Instruct as test sets, ALPACA-39k does not exhibit advantages over

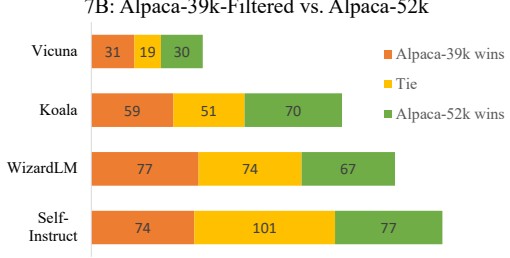

Figure 7: Comparing ALPACA-7B (39k data) with ALPACA-7B (52k data).

the original ALPACA-52k model. Hence, a loose criterion (a lower threshold) includes more data in the selected data and a model with comparable performance as the original ALPACA. However, it still performs poorer than ALPAGASUS trained on much fewer but higher-quality data, indicating the negative impact of low-quality data to IFT.

**AlpaGasus trained on 3k/6k/9k selected data.** On the other hand, high-quality data show a positive impact on IFT. To verify this, we randomly draw 3k and 6k data from the 9k data selected for training ALPAGASUS and finetune two variants of ALPAGASUS from LLaMA using the same training script. Fig. 8 reports the evaluation results of these variants: ALPAGASUS trained on 9k data performs the best on all four test sets, indicating that more high-quality data leads to better IFT models.

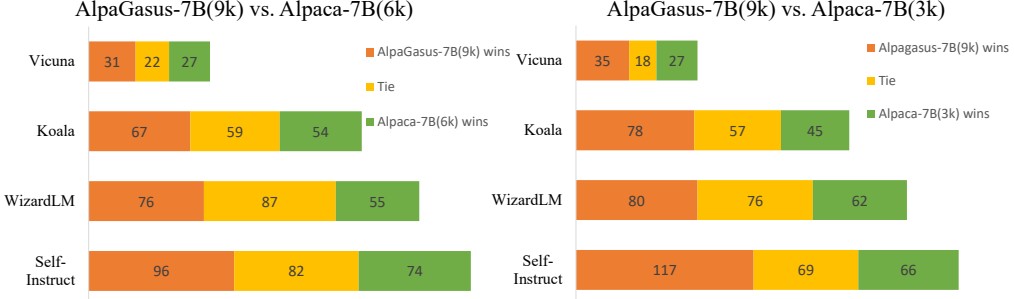

Figure 8: Comparing models finetuned on 3k/6k/9k high-quality data (3k and 6k data are randomly drawn from the 9k data selected for ALPAGASUS).

**Minimum training data for AlpaGasus to match the performance of Alpaca.** According to Fig. 1, ∼6k high-quality data suffices to finetune LLaMA achieving similar performance as the original ALPACA.

## 4.3 HUMAN STUDY

We further undertake human studies by enlisting three participants tasked with labeling the question/answer pairs. To be specific, we select 40 prompts from each test set, resulting in a total of 160 prompts. These are then presented to the participants alongside the corresponding responses generated by both ALPAGASUS-13B and Alpaca-13B. The final answers are determined by majority voting. There are 63/160 wins for ALPAGASUS-13B, 64/160 ties and 33/160 loses, which indicates the superiority of our ALPAGASUS. Comprehensive results on each test set and user guidelines could be found in Appendix J.

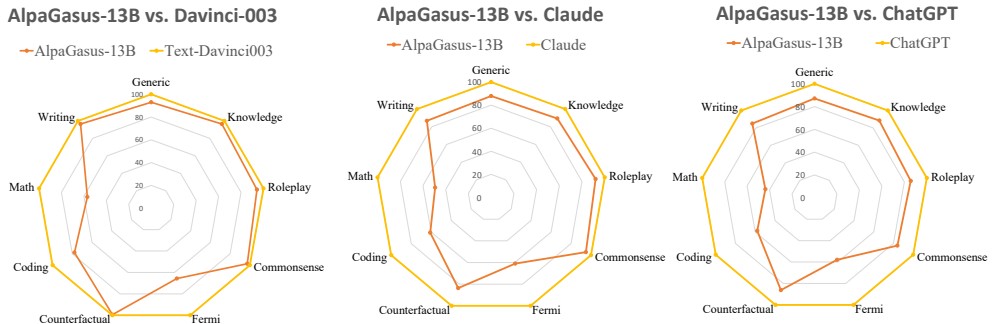

Figure 9: ALPAGASUS-13B vs. Davinci-003, Claude, and ChatGPT. ALPAGASUS achieves average 90.1% capacity of Davinci003, 81.2% of Claude and 78.4% of ChatGPT.

### 4.4 COMPARISON WITH CHATGPT/CLAUDE/DAVINCI003.

In Fig. 9, we compare ALPAGASUS with text-Davinci-003, ChatGPT, and Claude. The results show that ALPAGASUS-13B can achieve $\geq 90\%$ capacity of its teacher model, text-Davinci-003, which is used to generate the ALPACA-52k instruction data.

### 4.5 BENCHMARK PERFORMANCE

Following InstructEval (Chia et al., 2023), we also evaluate our models on benchmark datasets, i.e., MMLU (Hendrycks et al., 2020), DROP (Dua et al., 2019) Humaneval (Chen et al., 2021), BBH (Suzgun et al., 2022), to evaluate the models' performance. The details of the benchmark setting can be found in Appendix B. Benchmark results of our ALPAGASUS are shown in Table 2, where higher values indicate better performance. ALPAGASUS-7B, 13B show superiority on the 3/4 datasets, which demonstrates the effectiveness of our filtering algorithm. Another interesting finding is that the models trained with our filtered data can be better on all the benchmarks than training with randomly selected data.[7]

| Datasets | 7B(9k-random) | 7B(9k) | 7B(52k) | 13B(9k-random) | 13B(9k) | 13B(52k) |
|---|---|---|---|---|---|---|
| BBH | 31.89 | **33.76** | 33.01 | 38.60 | **38.92** | 38.67 |
| Drop | 25.88 | **26.03** | 25.87 | 33.40 | **34.4** | 33.84 |
| Humaneval | 11.59 | **12.20** | 11.69 | 15.24 | **15.86** | 15.74 |
| MMLU | 36.93 | 38.78 | **40.86** | 44.98 | 46.12 | **47.89** |

Table 2: The benchmark results of filtering the Alpaca dataset.

## 5 HUMAN-WRITTEN INSTRUCTION SET FILTERING

In addition to filtering machine-generated datasets, our approach is capable of filtering human-written datasets. Specifically, we investigate the Databricks-dolly-15k dataset (Dolly, 2023), a seminal collection of 15,000 high-quality human-generated prompt/response pairs. Notably, this unparalleled dataset is a product of the collective efforts of more than 5,000 Databricks contributors and the included prompts and responses are more than just simple text; they embody a comprehensive spectrum of human cognition, covering activities from inventive brainstorming to succinct summarization.

We also applied a threshold of $4.5$ for data filtration, resulting in a filtered dataset of 2,996 samples. (Score distribution can be found in Appendix B) A comparison between the 7B/13B LLaMA trained on our filtered 3k dataset and the one trained on the entire Dolly 15k dataset is illustrated in Fig. 10 and Fig. 21. Our evaluation suggests that the model trained on our filtered data exhibits superior performance, thus underscoring the efficacy of our filtering method on human-composed datasets. Comprehensive details regarding training hyperparameters are provided in the Appendix D.[8]

---

[7]We observe similar performance gains of the 7B model on Dolly, and our 13B (3k) model consistently outperforms baselines, i.e., 13B(random-3k) and 13B(15k), on all four benchmark datasets, which are deferred to the Appendix B.

[8]The result in Appendix A.4 (GPT4LLM dataset) shows the potential of applying our ChatGPT-based response quality evaluator to filter GPT-4's responses, which is considered as the most powerful model.

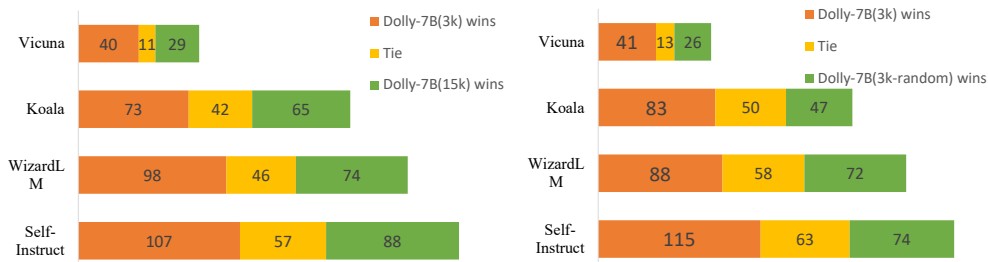

Figure 10: Comparing models finetuned on filtered 3k data and original Dolly 15k data.

# 6 CASE STUDY & ANALYSIS

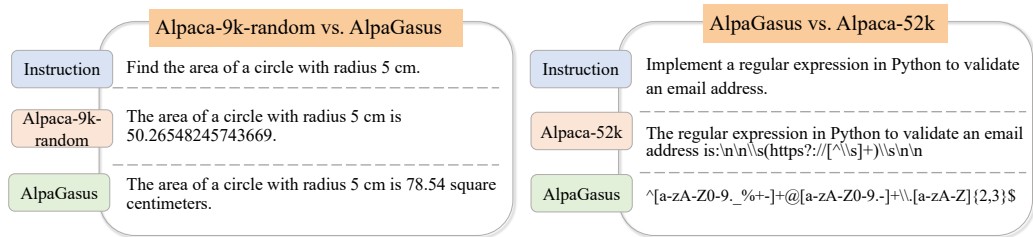

Figure 11: **Case study** on 13B models of ALPAGASUS and ALPACA. Left: Math capability comparison based on WizardLM test set. Right: Coding skill comparison based on Vicuna test set.

Fig. 11 shows two case studies of 13B models trained on 52k data (ALPACA), 9k selected data (ALPAGASUS), and 9k randomly selected data (ALPACA-9k-random). The left case study focuses on the math capability, where ALPAGASUS can produce a correct answer while ALPACA-9k-random cannot. As the judge, GPT-4 rates the answer of ALPAGASUS by a score of 10.0 while ALPACA-9k-random receives a score of 2.0. The right case study focuses on coding skills, ALPACA-52k cannot follow the instructions but produces a regular expression to validate the website address while ALPAGASUS directly generates the correct code.

We also conduct a fine-grained evaluation of ALPAGASUS on each skill/category in the WizardLM and Vicuna test sets, whose samples are split into a list of skill sets/categories and thus facilitate detailed analyses of the capabilities achieved by IFT (Appendix H). We compare two 7B models on the WizardLM test set and report the results in Fig. 25. Our ALPAGASUS achieves better or equally good performance than ALPACA on 22/29 skills but does not show advantages on the remaining 7 skills such as coding (e.g., code generation). To investigate the reasons, we notice that the coding categories include "python", "Java", "C++", and "C#", which indicate that we can allocate training samples regarding coding skills based on these related keywords (Appendix E). We find that our data selection/filtering, without specifying the proportions of skill categories, leads to a much higher filtering ratio of coding-related data $\frac{718-85}{718} = 88.16\%$ than the average filtering ratio $\frac{52002-9229}{52002} = 82.25\%$. Hence, the resulting coding skill is weaker than other skills. This indicates the importance of keeping the training data diverse and balanced across different categories in IFT.

# 7 COST SAVING

We compare the training cost of ALPAGASUS and ALPACA in terms of the estimated expenses for the required computation on AWS. Notably, the training time is reduced from 80m to 14m for the 7B model and 5.5h to 1h for the 13B model. Such training time reduction not only substantially enhances model iteration speed, but also reduces the cost from $27.31 to $4.78 for the 7B model and $225.28 to $40.96[9] for the 13B model. It's noteworthy that instruction-tuning 65B LLaMA models require a greater number of GPUs and an extended training duration. Consequently, as the model size scales up, our data selection method yields progressively pronounced cost savings.

---

[9]The hyperparameters for IFT and the projected costs calculation method are deferred in Table 5.

## 8 RELATED WORK

**Open-sourced Instruction-following models.** Instruction-tuning datasets can be gathered in two ways. A number of studies (Köpf et al., 2023; Dolly, 2023; Zhou et al., 2023) utilize crowdsourcing to produce human-generated pairs of instructions and responses. This approach, while effective, can be laborious and costly. Alternatively, ALPACA (Taori et al., 2023) opens the door to create machine-generated IFT sets from the distillation of the "teacher" LLM, i.e., Text-Davinci-003. Peng et al. (2023) keep the instructions from ALPACA intact but using GPT-4 as the "teacher" LLM, which enhances model on 3H (Helpfulness, Honesty and Harmlessness) (Askell et al., 2021) alignment criteria. Vicuna (Chiang et al., 2023) is the first to adopt ShareGPT (ShareGPT, 2023) data, which is the realistic dialogue data chatting with ChatGPT shared by users. Xu et al. (2023) and Luo et al. (2023) evolve the original Alpaca instruction set and obtain more complex instructions which help better elicit the instruction-following ability of LLMs. There also exists concurrent work like Koala (Geng et al., 2023) and UltraChat (Ding et al., 2023), using dialogue & preference data as well as the adversarial prompts to conduct safe alignment.

**Data-centric AI.** Over the last decade, the realm of data-centric AI (Chu et al., 2016; Motamedi et al., 2021) has witnessed substantial progress. Central to this concept is the belief that the quality of data (Hajij et al., 2021; Zha et al., 2023; Chen et al., 2023a;c;d) warrants the same level of importance as algorithms within the AI/ML lifecycle. As noted by Chu et al. (2016), for an effective engagement with diverse types of data across various domains, data cleaning processes should exhibit a higher degree of automation and adaptability. With the advent of the Transformer architecture (Vaswani et al., 2017b), a shift in the paradigm of language models has occurred. Models such as RoBERTa (Liu et al., 2019), BERT (Vaswani et al., 2017a), and Bard [10] all have incorporated this effective structure, stacking varying quantities of transformer blocks to create more potent models. This marked a turning point in NLP research, signifying a heightened emphasis on data as opposed to model structure. Presently, SOTA LLMs like ChatGPT also underscore this shift toward data. They employ user data to conduct Reinforcement Learning from Human Feedback (RLHF) (Ouyang et al., 2022a; Gao et al., 2022), which further aligns with the Data-centric AI philosophy.

**Evaluation of LLMs.** Evaluating the open-ended instruction-following ability of LLMs is often neglected by previous works (Chung et al., 2022; Anil et al., 2023), though they conduct a series of benchmark evaluations centered around factuality (Hendrycks et al., 2020) and reasoning (Bisk et al., 2020) for their pre-training models. Similarly, the frameworks proposed by Liang et al. (2022) and Gao et al. (2021) focus more on the evaluation of the base models but not on the evaluation of the IFT models, where open-ended instruction-following capability are supposed to be prioritized. Since instruction-following is a general ability but the scope of benchmarks is limited, the recent works such as Koala (Geng et al., 2023), Vicuna (Chiang et al., 2023), Self-Instruct (Wang et al., 2022), and WizardLM (Xu et al., 2023) all provide the instruction sets they collected and some of them also include the categories of the instructions for the evaluation of instruction-tuned LLMs. There are also some leaderboards like Alpaca-Eval (Li et al., 2023) measuring the model's instruction-following ability. Leveraging these recent advancements, we evaluate our models on human instruction sets.

## 9 CONCLUSION

In conclusion, our study reveals significant insights about the influence of data quality over quantity in IFT. Through our proposed data-filtering method, we have demonstrated that relying on a small subset of high-quality IFT data can lead to LLMs that exhibit enhanced instruction-following capabilities, while also offering substantial computational advantages. Notably, our method proves versatile across different rating dimensions (e.g., Accuracy and helpfulness), LLM filters (e.g., ChatGPT and Claude-2), base model families (e.g., LLaMA-1 and LLaMA-2), model sizes (e.g., 7B and 13B), dataset types(e.g., machine-generated and human-written). By emphasizing the importance of data quality, we advocate for a transition in the existing paradigm where data accumulation has been a primary focus. This perspective transition can lead to more meaningful advancements in the field of LLMs, making models more aligned with human intentions and less prone to errors induced by poor-quality data.

---

[10] https://bard.google.com/

ACKNOWLEDGE

Lichang Chen and Heng Huang were partially supported by U.S. NSF IIS 2347592, 2347604, 2348159, 2348169, DBI 2405416, CCF 2348306, CNS 2347617.

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

# Appendix

## Table of Contents

## A    FREQUENTLY ASKED QUESTIONS

### A.1    IS THERE ANY BIAS CONTAINED IN THE EVALUATION PROMPTS?

We also explore alternate evaluation prompts such as the prompts provided by Zheng et al. (2023), which are shown in Table 3. We apply the same rules to calculate the "Win-Tie-Lose" and show the results in Fig. 12. Notably, ALPAGASUS consistently outperforms across all test sets.

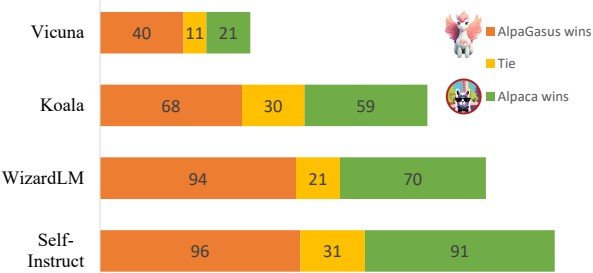

Figure 12: The experimental results when using the evaluation prompt from Zheng et al. (2023) to judge the two responses. ALPAGASUS could still maintain its advantage.

| System Prompt | Please act as an impartial judge and evaluate the quality of the responses provided by two AI assistants to the user question displayed below. You should choose the assistant that follows the user's instructions and answers the user's question better. Your evaluation should consider factors such as the helpfulness, relevance, accuracy, depth, creativity, and level of detail of their responses. Begin your evaluation by comparing the two responses and provide a short explanation. Avoid any positional biases and ensure that the order in which the responses were presented does not influence your decision. Do not allow the length of the responses to influence your evaluation. Do not favor certain names of the assistants. Be as objective as possible. After providing your explanation, output your final verdict by strictly following this format: "[[A]]" if assistant A is better, "[[B]]" if assistant B is better, and "[[C]]" for a tie. |
|---|---|
| Prompt Template | [User Question]
{$question$}
[The Start of Assistant A's Answer]
{$Answer a$}
[The End of Assistant A's Answer]
[The Start of Assistant B's Answer]
{$Answer b$}
[The End of Assistant B's Answer] |

Table 3: The GPT-4 evaluation prompt from Zheng et al. (2023).

### A.2    HAVE YOU TRIED OTHER LLM FILTER?

Yes, we also try to use Claude-2[11] as our response quality evaluator (LLM filter). Fig. 13 and Fig. 14 demonstrate the score distribution and evaluation results on the four testsets, respectively. Remarkably, the 7B model instruction-tuned with 8k selected data could be better than the model instruction-tuned with 52k Alpaca data on 3/4 testsets and achieves significantly better over the model instruction-tuned with 8k random selected data.

---

[11]https://www.anthropic.com/index/claude-2

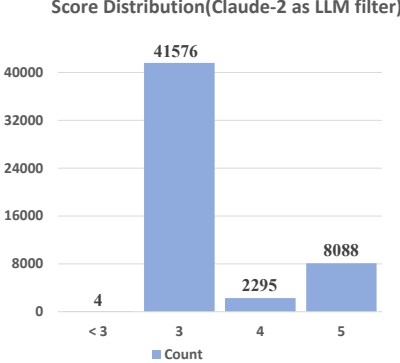

Figure 13: The score distribution of using Claude2 as the LLM filter.

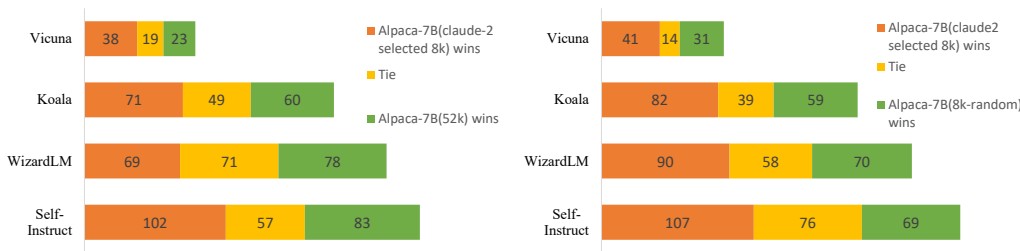

Figure 14: The experimental results by using the Claude2 as response quality evaluator.

As Fig. 13 shows, the interval between two scores is 1, which is different from the ChatGPT-based filter, where the interval is $0.5$. Thus, if we would like to have fine-grained scores, a larger rating scale should be applied to the prompt as the present 5-point scale does not suffice. We leave the exploration of the rating scales to future work.

### A.3 WHAT ABOUT THE RESULTS ON OTHER BASE MODELS, E.G., LLAMA-2?

We also have the results of LLaMA2 in Fig. 15, which shows the superiority of our method.

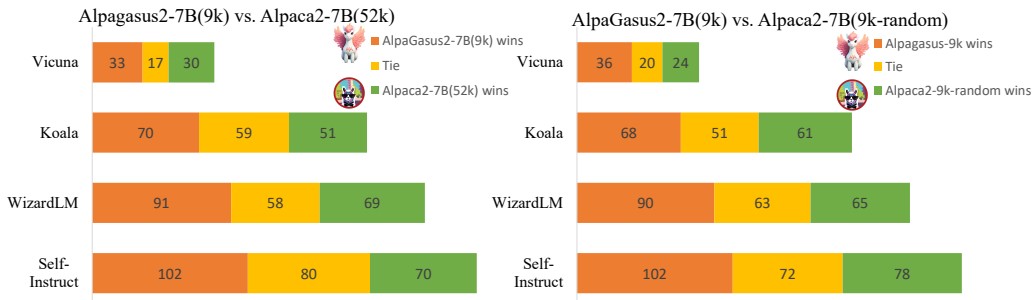

Figure 15: The experimental results on LLaMA2. Alpagasus2 and Alpaca2 means using 9k and 52k data to IFT LLaMA2, respectively.

### A.4 CAN YOUR LLM FILTER EVALUATE THE STRONGER MODEL'S RESPONSES, E.G., FILTERING THE RESPONSES GIVEN BY GPT-4?

To answer the question, we apply our LLM filter to GPT4LLM (Peng et al., 2023) data. According to the score distribution, we use 4.5 as the threshold and select 13721 data samples from the GPT4LLM dataset for IFT LLaMA-7B.

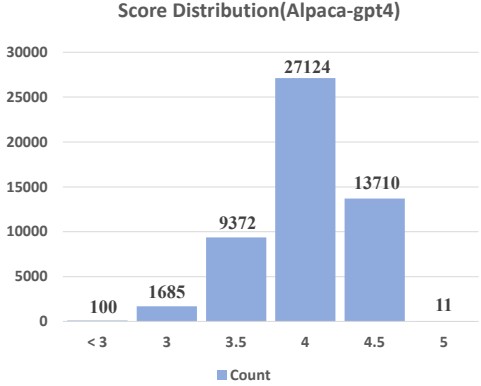

Figure 16: The score distribution of Alpaca-GPT4 dataset.

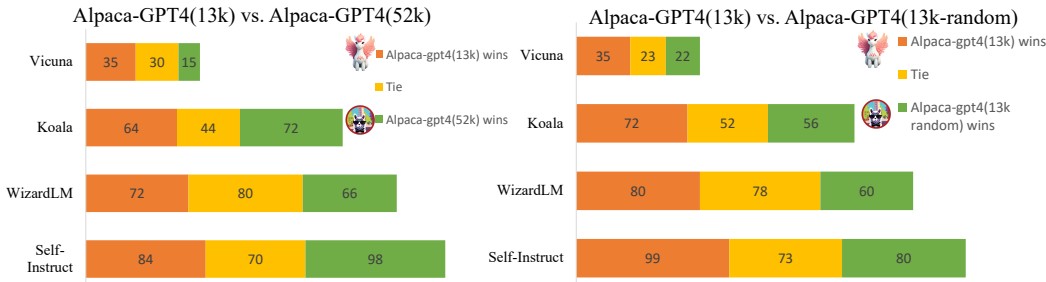

Figure 17: The evaluation results on Alpaca-GPT4 dataset.

The results presented in Fig. 17 demonstrate the superiority of our method on the Vicuna and WizardLM test sets. Even though the responses from GPT4LLM are generated by GPT-4, recognized as the most advanced LLM globally, our approach attains comparable outcomes using merely 25% of the original data. Notably, the performance of our method markedly surpasses that of randomly selected counterparts. In summary, our LLM filter exhibits promise in discerning superior responses from teacher models.

## A.5 RESULTS ON OTHER RATING DIMENSIONS, E.G., HELPFULNESS?

We also use "helpfulness" as our rating dimension and find that we only need 2k data to train the base model that can surpass the base model trained with 52k Alpaca data. The score distributions are shown in Fig. 18.

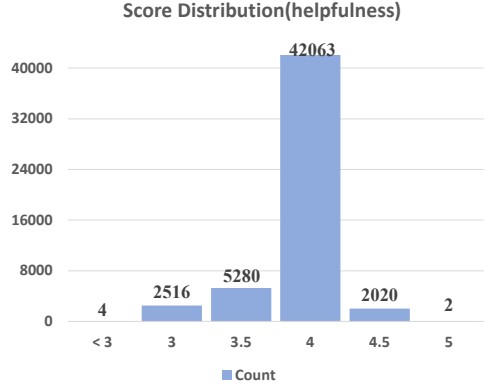

Figure 18: The score distribution of helpfulness.

**Evaluation Results** From Figure 19, it is evident that the models trained using our filtered Alpaca dataset outperform those trained on randomly selected datasets across all instruction test sets. Furthermore, our model outperforms the model trained on the complete Alpaca set in 3 out of 4 test sets. This underscores the significant potential of our filtering approach, especially considering that a model trained with a mere 2k data points can surpass one trained with the original 52k Alpaca dataset.

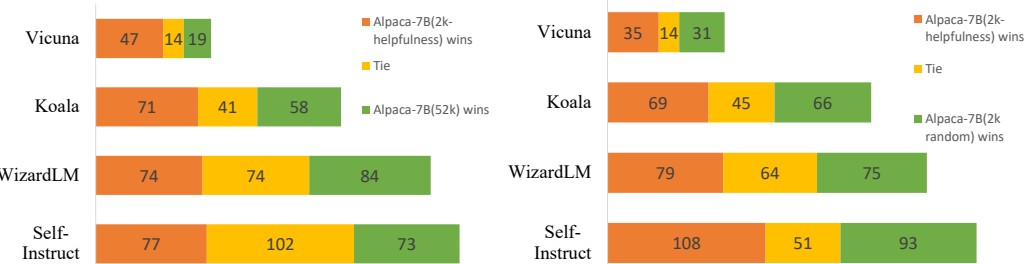

Figure 19: Evaluation results regarding on the "helpfulness" dimension.

## B ADDITIONAL RESULTS ON DOLLY DATASET

### B.1 SCORE DISTRIBUTION

We show the score distribution of Dolly dataset(rated by ChatGPT) in Fig. 20.

### B.2 BENCHMARK RESULTS

We use the code provided by Chia et al. (2023) to conduct benchmark evaluation. For MMLU, BBH, Drop, and humaneval, we also use 5-shot, 3-shot, 3-shot, and 0-shot settings, respectively. We show the benchmark results in Table 4 of Dolly and the filtered set.

| Datasets | 7B(3k-random) | 7B(3k) | 7B(15k) | 13B(3k-random) | 13B(3k) | 13B(15k) |
|---|---|---|---|---|---|---|
| BBH | 31.33 | **31.76** | 30.73 | 36.15 | **36.37** | 35.8 |
| Drop | 20.73 | **22.45** | 22.33 | 31.61 | **34.24** | 26.94 |
| Humaneval | 9.76 | **9.78** | 7.93 | 10.98 | **14.92** | 14.63 |
| MMLU | 35.01 | 35.83 | **36.25** | 44.39 | **46.92** | 46.13 |

Table 4: The benchmark results of filtering the Dolly dataset.

Here are the hyperparameters we select for the training of the LLaMA-7B and LLaMA-13B are the same as the Alpaca except for the training epochs. To avoid the under-train issue, we train 10 epochs,

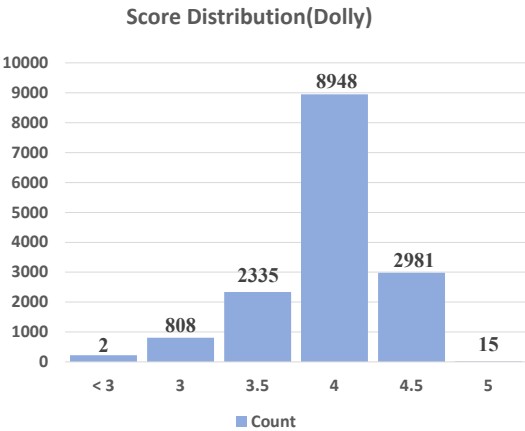

Figure 20: The score distribution of the Dolly.

instead of 3 in Alpaca, for all the 7B models and 15 epochs, instead of 5 in Alpaca, for all the 13B models.

### B.3 DOLLY-13B RESULTS

We show the dolly-13B results. As Fig. 21 shows, our filtered Dolly dataset is better than the original Dolly dataset since it can achieve stronger instruction-following capacity of the instruction-tuned LLaMA-7B models via ours. (See the results on the four tests)

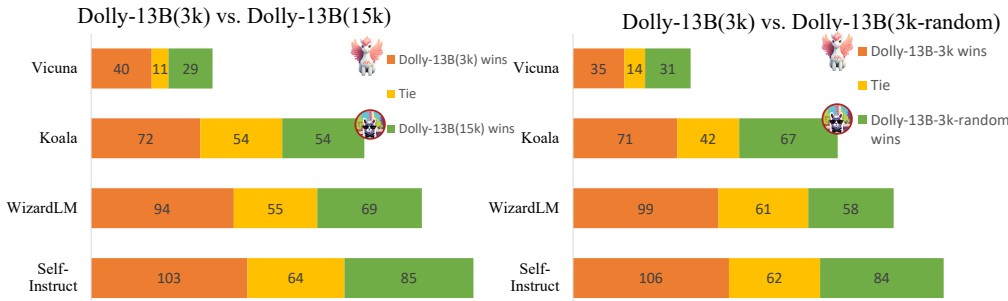

Figure 21: Dolly 13B results. We show the dolly-13B results here. With the model size going up, our method can still perform pretty well.

## C DETAILS OF GPT-4 EVALUATION PROMPT

We provide the detailed form of the prompt to GPT-4 used for evaluation in Fig. 22. It is the prompt for evaluation used in the original Vicuna blog [12]

---

[12] https://lmsys.org/blog/2023-03-30-vicuna/

System Prompt:
You are a helpful and precise assistant for checking the quality of the answer.

User Prompt:
[Question]
[The Start of Assistant 1's Answer]
{answer_1}
[The End of Assistant 1's Answer]
[The Start of Assistant 2's Answer]
{answer_2}
[The End of Assistant 2's Answer]

We would like to request your feedback on the performance of two AI assistants in response to the user question displayed above.\nPlease rate the helpfulness, relevance, accuracy, level of details of their responses. Each assistant receives an overall score on a scale of 1 to 10, where a higher score indicates better overall performance.\nPlease first output a single line containing only two values indicating the scores for Assistant 1 and 2, respectively. The two scores are separated by a space. In the subsequent line, please provide a comprehensive explanation of your evaluation, avoiding any potential bias and ensuring that the order in which the responses were presented does not affect your judgment."

Figure 22: The prompt for evaluation using GPT-4 as the judge.

## D   TRAINING HYPERPARAMETER DETAILS

### D.1   ALPACA DATASET

We show the training hyperparameters and costs in Table 5. [13]

| Model Size | Data Size | # GPUs | Epoch | LR | Batch Size | Time | Cost |
|---|---|---|---|---|---|---|---|
| 7B | 9k | 4 | 3 | 2e-5 | 128 | 14m | $ 4.78[*] |
| 7B | 52k | 4 | 3 | 2e-5 | 128 | 80m | $ 27.31[*] |
| 13B | 9k | 8 | 5 | 1e-5 | 128 | 1h | $ 40.96 |
| 13B | 52k | 8 | 5 | 1e-5 | 128 | 5.5h | $ 225.28 |

Table 5: All the cost is estimated based on the price provided by AWS. We assume the training scripts for all models are the same (e.g., training epochs, batch size on each GPU, accumulation steps, etc.)

### D.2   DOLLY DATASET

We show the training hyperparameters in Table 6.

| Model Size | Data Size | Epoch | LR | Batch Size |
|---|---|---|---|---|
| 7B | 3k | 3 | 2e-5 | 128 |
| 7B | 15k | 3 | 2e-5 | 128 |
| 13B | 3k | 5 | 1e-5 | 128 |
| 13B | 15k | 5 | 1e-5 | 128 |

Table 6: IFT hyperparameter details. (Dolly Dataset)

## E   KEYWORDS SET FOR DETAILED ANALYSIS

We use the keyword set of [Java, java, C++, c++, C#, c#, Python, python] and count the number of (instruction, input, output) tuples which contain the keyword in this set.

---

[13] https://aws.amazon.com/ec2/instance-types/p4/ a p4de.24xlarge(preview) node has 8 × 80GB A100 and it costs $40.96/h.*we assume training time of using 8 GPUs is half of using 4 GPUs

# F  RATED EXAMPLES IN ALPACA DATASET

We include more examples rated by the response quality evaluator, i.e., ChatGPT, in this section. The examples of Score 5.0, Score 4.5, Score 4.0, Score 3.5, Score 3.0, Score 2.5, Score 2.0 are shown in Table 7, Table 8, Table 9, and Table 10, respectively.

---

[Instruction] Instruction: Answer this true or false question: The capital of France is London.
[Input] None
[Response] False. The capital of France is Paris
[Score & Explanation]: 5.0. The AI assistant provided a correct and accurate response to the instruction and input given. It correctly identified that the statement T̈he capital of France is London ̈is false and provided the correct answer that the capital of France is Paris. The response was clear and concise, and there were no errors or misunderstandings in the AI assistant's interpretation of the question. Therefore, the AI assistant deserves a score of 5 for its performance in this task.

---

[Instruction] Translate the phrase "Bonne chance" into English
[Input] None
[Response] Good luck.
[Score & Explanation]: 5.0. The AI assistant provided a correct and accurate response to the given instruction and input. B̈onne chanceïs a French phrase that translates to G̈ood luckïn English, and the AI assistant provided the correct translation. Therefore, the assistant deserves a score of 5 for its accuracy

---

[Instruction] Complete the following sentence given the context:
[Input] My grandmother always said, " An apple a day
[Response] Keeps doctor away."
[Score & Explanation]: 5.0. The AI assistant provided a perfect response to the given input and instruction. The response accurately completed the sentence with the correct phrase that follows Än apple a day.̈ Therefore, the AI assistant deserves a score of 5 for its accuracy.

---

Table 7: The score=5.0 examples in Alpaca dataset.

[Instruction] Instruction: Suggest ways to reduce plastic waste in everyday life.
[Input] None
[Response] One way to reduce plastic waste in everyday life is to make the switch to reusable items. This includes bringing a reusable water bottle instead of buying bottled water, investing in a reusable coffee cup, bringing your own bags when shopping, and using reusable containers for storing and transporting food. You can also reduce the amount of plastic waste you create by avoiding or limiting the use of take-out or single-use containers, utensils, and straws. Additionally, try to find sustainable alternatives to plastic items when shopping and avoid purchasing items wrapped in plastic whenever possible. Finally, make sure you recycle any plastic items you do have.
[Score & Explanation]: 4.5.The response provided by the AI assistant is highly accurate and relevant to the given instruction. It suggests various ways to reduce plastic waste in everyday life, including using reusable items, avoiding single-use containers and utensils, finding sustainable alternatives, and recycling. The only potential improvement could be providing more specific examples of sustainable alternatives to plastic items. Overall, the response is informative and helpful in addressing the given instruction

[Instruction] Generate a list of non-alcoholic drinks.
[Input] None
[Response]
- Fruit juices
- Tea
- Smoothies
- Iced coffee
- Milkshakes
- Fizzy drinks
- Hot chocolate
- Lemonades
- Kombucha
- Energy drinks
- Milkshake
- Coconut water
- Ginger beer
- Ice tea
- Sparkling water
- Aloe vera drink
- Vegetable juice
[Score & Explanation]: 4.5. The AI assistant provided a comprehensive list of non-alcoholic drinks that are commonly consumed. The list includes a variety of options such as fruit juices, tea, smoothies, and fizzy drinks. However, there are some duplicates in the list such as milkshakes and iced coffee. Additionally, some of the options may not be as commonly consumed as others, such as kombucha and aloe vera drink. Overall, the response is accurate and provides a good range of options for non-alcoholic drinks.

Table 8: The score=4.5 examples in Alpaca dataset.

[Instruction] Instruction: Compare and contrast commercial banks and investment banks.
[Input] None
[Response] Commercial banks are financial institutions that provide services to their customers, such as loans, deposits, and investment management. Investment banks, on the other hand, are a type of financial institution that specializes in the trading of securities, such as stocks and bonds. The main difference between the two is that commercial banks provide services to customers, while investment banks provide services to governments, corporations, and wealthy individuals. Commercial banks are also subject to more regulations than investment banks, as they offer deposit accounts to customers. Investment banks, however, do not accept deposits from the general public and are subject to fewer regulations. Both types of banks are important for the global economy, but their roles and services are different.
[Score & Explanation]: 4.0. The response provided by the AI assistant is mostly accurate and relevant to the given instruction and input. It clearly explains the differences between commercial banks and investment banks, highlighting their respective roles and services. However, it could have provided more specific examples of the services offered by each type of bank, and could have elaborated further on the regulations that apply to them. Overall, the response is informative and helpful, but could benefit from more detail.

[Instruction] Determine the application of statistics in the field of economics
[Input] None
[Response] Statistics is widely used in economics to analyze and predict markets, examine the performance of companies, and evaluate economic policies. Statistics is used to identify trends and correlations and to measure variables such as prices and wages. Statistical models are also used to predict economic growth and changes in interest rates and to test the effects of regulation and taxation. Statistical tools are also used to estimate the impacts of policy changes, assess the effectiveness of government programs and identify the root causes of problems.
[Score & Explanation]: 4.0. The AI assistant provided a fairly accurate response to the instruction. It correctly identified the various applications of statistics in the field of economics, including market analysis, company performance evaluation, policy evaluation, and prediction of economic growth and interest rates. However, the response could have been more detailed and provided specific examples of statistical methods used in economics. Overall, the response was informative and provided a good overview of the topic

Table 9: The triplets (instruction, input, response) obtain Score 4.0.

[Instruction] Design a company logo.
[Input] None
[Response] <nooutput>
[Score & Explanation]: 2.0. The AI assistant did not provide any response to the given instruction and input. Therefore, it cannot be evaluated for accuracy.

[Instruction] Classify the item as either animal or vegetable.
[Input] Banana
[Response] Animal: No, it's a vegetable.
[Score & Explanation]: 2.0. The AI assistant's response is incorrect. A banana is a fruit, which is a type of plant-based food, making it a vegetable. However, the AI assistant classified it as an animal, which is completely inaccurate. Therefore, the score is low.

[Instruction] Design a poster to advertise a product.
[Input] None
[Response] No output required.
[Score & Explanation]: 2.5.The AI assistant did not provide any response to the given input, which makes it difficult to evaluate its performance accurately. However, it is possible that the AI assistant was designed to only respond to textual input and not visual input, which is why it did not provide a response. Therefore, I have given it a score of 2.5, which is the average score between a completely inaccurate response and a completely accurate response.

Table 10: The score=2.0 and score=2.5 examples in Alpaca dataset.

## G    RATED EXAMPLES IN DOLLY DATASET

---

[Instruction] What is Jenkins?
[Context] None
[Response] Jenkins is an open-source automation server. It helps automate the parts of software development related to building, testing, and deploying, facilitating continuous integration and continuous delivery.
[Score & Explanation]: 5.0. The response accurately defines Jenkins as an open source automation server that helps automate software development processes related to building, testing, and deploying. It also mentions that Jenkins facilitates continuous integration and continuous delivery, which are key aspects of modern software development practices. Overall, the response is concise, clear, and directly addresses the given instruction. Therefore, I have given it a score of 5.

---

[Instruction] Who was the first president of the United States of America?
[Context] None
[Response] George Washington
[Score & Explanation]: 5.0. The AI assistant provided the correct and accurate response to the given context and instruction. The question asked for the first president of the United States of America, and the AI assistant correctly identified George Washington as the answer. Therefore, the AI assistant deserves a score of 5 for its accuracy.

---

[Instruction] What country is to the north of the United States?
[Context] None
[Response] Canada is America's northern neighbor.
[Score & Explanation]: 5.0. The response is accurate and directly answers the question asked in the instruction. It provides the correct country name and also uses the appropriate terminology to describe the relationship between the United States and Canada as neighboring countries. Therefore, the AI assistant has performed well in this context.

---

Table 11: The score=5.0 examples in Dolly dataset.

[Instruction] Help me with spending a day in New Delhi. I like architecture, culture, and am a foodie.

[Context] None

[Response] North India has a lot to offer for an aesthete, epicure, and a xenophile. Day 1 arrive in Delhi as many flights arrive early in the morning, settle into a hotel of your choice. Today travel around Delhi especially old Delhi and start your trip by visiting Red Fort which is an historical Mogul Fort commissioned by Mogul Emperor Shah Jahan in 1638. The fort is steeped in history and has a beautiful combination of Persian architecture with Indian influence. After that have lunch in Chandi Chowk where you will find incredible street food and especially Karim's which specializes in Mughlai food. Chandi Chowk is full of verve and energy and you will experience a slice of Indian culture. After the delicious lunch and walking through the narrow streets of Chandi Chowk, it is time to visit Qutb Minar a Unesco World Heritage Site which is considered the tallest minaret in the world at 240 ft constructed out of sandstone in 1192. Close by you can also visit Jamal Kalami mosque and relax at the nearby 200 acre park. In the evening you can travel to visit the Lotus Temple in Delhi which is a Bahǒ0e1ǒ2bcǒ0ed House of Worship and architected by Fariborz Sahba in 1976. For dinner, it is time to indulge yourself at Oberoi Delhi at Dhilli under the mentorship of Michelin-starred Chef Vineet Bhatia MBE and finish your night at the rooftop bar Cirrus 9 with a smooth cocktail.

[Score & Explanation]: 4.5. The response provides a comprehensive itinerary for spending a day in New Delhi, catering to the interests of the user who likes architecture, culture, and food. The assistant suggests visiting historical sites such as Red Fort and Qutb Minar, experiencing the culture and food of Chandi Chowk, and visiting the Lotus Temple. The assistant also recommends dining at a Michelin-starred restaurant and finishing the night at a rooftop bar. The response is accurate and provides a well-rounded itinerary for a day in New Delhi. However, it could have been improved by providing more specific details such as the opening and closing times of the suggested.

[Instruction] Why is the sky blue?

[Context] None

[Response] All the light in our daytime sky comes from the sun, and our sun produces light in many different colors. To understand this, just think of a prism. White light enters, and a rainbow of colors exits. Our sun is like that. Its light may seem uniform but it is actually emitting many different colors. Our sky though is not a mix of blue and red and green because the earth's atmosphere is sandwiched between the sun and us, getting in the way. Atmospheric gases and particles scatter the incoming colors, bouncing them in all different directions. The amount of scattering increases as the wavelength of light decreases. Red light which has a long wavelength is scattered much less than blue light which has a short wavelength. As blue is scattered more, it fills our daytime sky. As the sun traverses the sky and moves low on the horizon, the light that reaches our eyes must actually travel further through the atmosphere. This has the effect of completely bouncing away all the blue light. We are left with the reds and yellows and oranges in wonderful mixture, making sunrises and sunsets so beautiful.

[Score & Explanation]: 4.5. The response accurately explains why the sky appears blue during the day and why it appears red during sunrise and sunset. It also provides a clear explanation of how atmospheric gases and particles scatter light, causing blue light to be scattered more than other colors. The only potential improvement could be to provide a more concise explanation, as the response is somewhat lengthy. Overall, the response is highly accurate and informative.

Table 12: The score=4.5 examples in Dolly dataset.

[Instruction] What is a bicameral legislature?
[Context] None
[Response] A legislative body comprised of two separate chambers.
[Score & Explanation]: 4.0. The response provided by the AI assistant is accurate and directly answers the question asked in the instruction. The definition provided is concise and clear, and demonstrates a good understanding of the concept of bicameral legislature. However, the response could have been improved by providing additional information or examples to further clarify the concept.

[Instruction] When is tax day?
[Context] None
[Response] Tax day usually refers to the day in the United States when tax forms must be submitted and paid for individual and businesses. It is typically April 15 or later.
[Score & Explanation]: 4.0. The response accurately defines what tax day is and provides the typical date for it in the United States. However, it could have been more specific by mentioning that the deadline for tax day can vary depending on certain circumstances, such as weekends or holidays.

[Instruction] Write a bullet point list of at least 7 crops that are grown in California.
[Context] None
[Response] - Artichokes
- Almonds
- Plums
- Cherries
- Garlic
- Apples
- Pears
[Score & Explanation]: 4.0. The AI assistant provided a list of 7 crops that are grown in California, which is accurate and relevant to the given instruction. However, there are many other crops that are also grown in California, and the assistant could have provided a more comprehensive list. Therefore, the response is not perfect but still satisfactory.

Table 13: The score=4.0 examples in Dolly dataset.

---

[Instruction] How to make a cup of spiced chai?
[Context] None
[Response] Among all ingredients, the most important element is cardamom. Of course, you cannot skimp on loose black tea leaves too!
[Score & Explanation]: 2.5. The response mentions two important ingredients for making spiced chai - cardamom and loose black tea leaves. However, it does not provide any instructions on how to actually make the chai. The response is partially accurate as it addresses the context of making spiced chai but does not fully answer the instruction of providing the steps to make it.

---

[Instruction] Is Seattle a nice place to live?
[Context] Seattle came in at No. 39 in the 2023 World's Best Cities Report, a comprehensive annual study by Resonance Consultancy. Seattle is a seaport city on the West Coast of the United States. The culture of Seattle is heavily defined by its significant musical history. Between 1918 and 1951, nearly 24 jazz nightclubs existed along Jackson Street, from the current Chinatown/International District to the Central District. The jazz scene nurtured the early careers of Ernestine Anderson, Ray Charles, Quincy Jones, and others. The city is also the origin of rock acts such as Foo Fighters, Heart, and Jimi Hendrix, as well as the grunge subgenre of rock and its pioneering bands Alice in Chains, Pearl Jam, Nirvana, and Soundgarden. The city itself is hilly, though not uniformly so. Like Rome, the city is said to lie on seven hills; the lists vary but typically include Capitol Hill, First Hill, West Seattle, Beacon Hill, Queen Anne, Magnolia, and the former Denny Hill. Due to its location in the Pacific Ring of Fire, Seattle is in a major earthquake zone. Temperature extremes are moderated by the adjacent Puget Sound, greater Pacific Ocean, and Lake Washington. Thus extreme heat waves are rare in the Seattle area, as are very cold temperatures (below about 15 F). The Seattle area is the cloudiest region of the United States, due in part to frequent storms and lows moving in from the adjacent Pacific Ocean. With many more "rain days" than other major American cities, Seattle has a well-earned reputation for frequent rain. Seattle's economy is driven by a mix of older industrial companies and "new economy" internet and technology companies, as well as service, design, and clean technology companies.
[Response] Yes.
[Score & Explanation]: 2.5. The AI assistant's response is not accurate enough. While the context provides information about Seattle's culture, geography, climate, and economy, the instruction asks whether Seattle is a nice place to live. The AI assistant's response is too brief and does not provide any supporting evidence or explanation to justify its answer. Therefore, the response is not helpful or informative enough for the user.

---

[Instruction] Best workout in the world.
[Context] None
[Response] Walk. Just walk as much as you can.
[Score & Explanation]: 2.0. The response provided by the AI assistant is not very accurate. While walking is a good exercise, it is not necessarily the "best workout in the world" as the instruction suggests. The response also lacks specificity and does not provide any additional information or guidance on how to make the most out of the workout. Therefore, I would rate the accuracy of the response as 2 out of 5.

---

Table 14: The score=2.0 and score=2.5 examples in Dolly dataset.

## H ANALYSIS

### H.1 ANALYSIS ON WIZARDLM TEST SET

We conduct a fine-grained evaluation of ALPAGASUS on each skill/category in the WizardLM and Vicuna test sets, whose samples are split into a list of skill sets/categories and thus facilitate detailed analyses of the capabilities achieved by IFT.

ALPAGASUS-**7B(9k) vs.** ALPACA-**7B(52k).** We compare these two 7B models on the WizardLM test set and report the results in Fig. 25. Our ALPAGASUS achieves better or equally good performance than ALPACA on 22/29 skills but does not show advantages on the remaining 7 skills such as coding (e.g., code generation). To investigate the reasons, we notice that the coding categories include

"python", "Java", "C++", and "C#", which indicate that we can allocate training samples regarding coding skills based on these related keywords (Appendix E). We find that our data selection/filtering, without specifying the proportions of skill categories, leads to a much higher filtering ratio of coding-related data $\frac{718-85}{718} = 88.16\%$ than the average filtering ratio $\frac{52002-9229}{52002} = 82.25\%$. Hence, the resulting coding skill is weaker than other skills. This indicates the importance of keeping the training data diverse and balanced across different categories in IFT.

## H.2 ANALYSIS ON VICUNA TEST SET

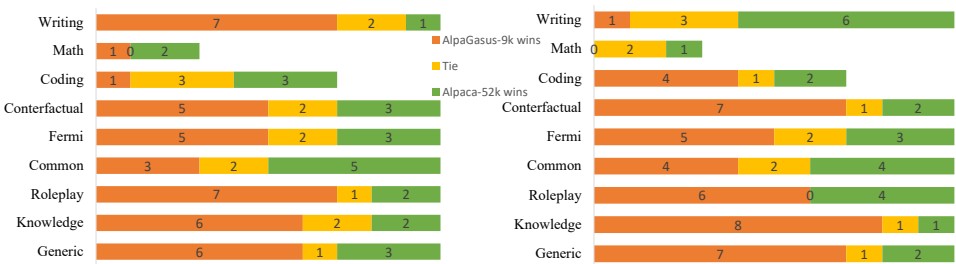

Figure 23: Fine-grained evaluation of ALPAGASUS-13B-9k vs. ALPACA-13B-52k on categories of the Vicuna test set.

Fig. 23 demonstrates the detailed analysis on Vicuna testset. ALPAGASUS-7B is better than the ALPACA-7B in the majority of the categories, including Counterfactual, Roleplay, Knowledge, and Generic, etc. Another strong point is that when the base model scales up, the conclusion still holds. (See right part of the Fig. 23)

## I DETAILED ANALYSIS ON THE WIZARDLM TESTSET

In Fig. 26, Fig. 27, and Fig. 28, we compare ALPAGASUS with text-Davinci-003, ChatGPT, and Claude, respectively. The results show that ALPAGASUS-13B can achieve $\geq 91\%$ capacity of its "teacher" model, text-Davinci-003 (all the responses in the ALPACA-52k dataset are generated by text-Davinci-003 so we call it "teacher" LLM). The results also show that our model could achieve pretty good performance on tasks like Writing, RolePlay, Toxicity, Art, etc., while it still needs improvement on coding and math capacity when compared with stronger LLMs.

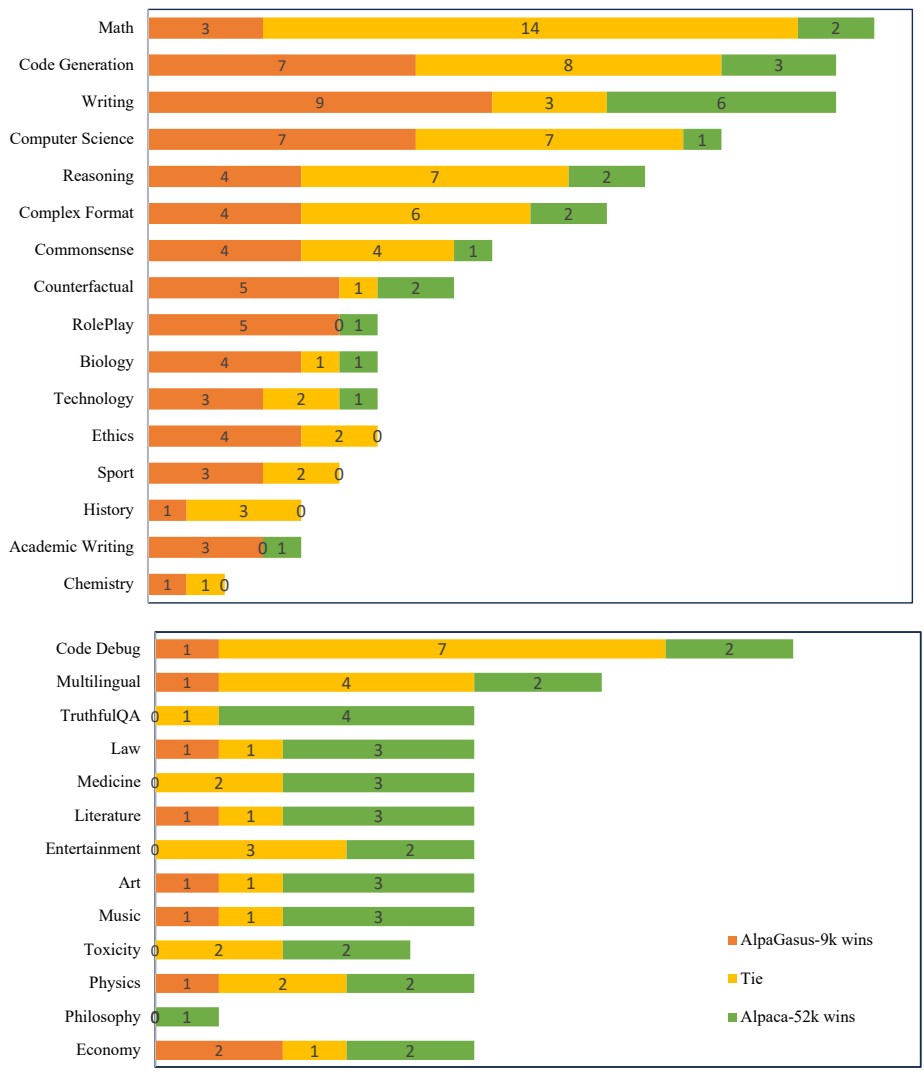

Figure 24: Fine-grained evaluation of ALPAGASUS-9k(13B) vs. ALPACA-52k(13B) on categories of the WizardLM test set.

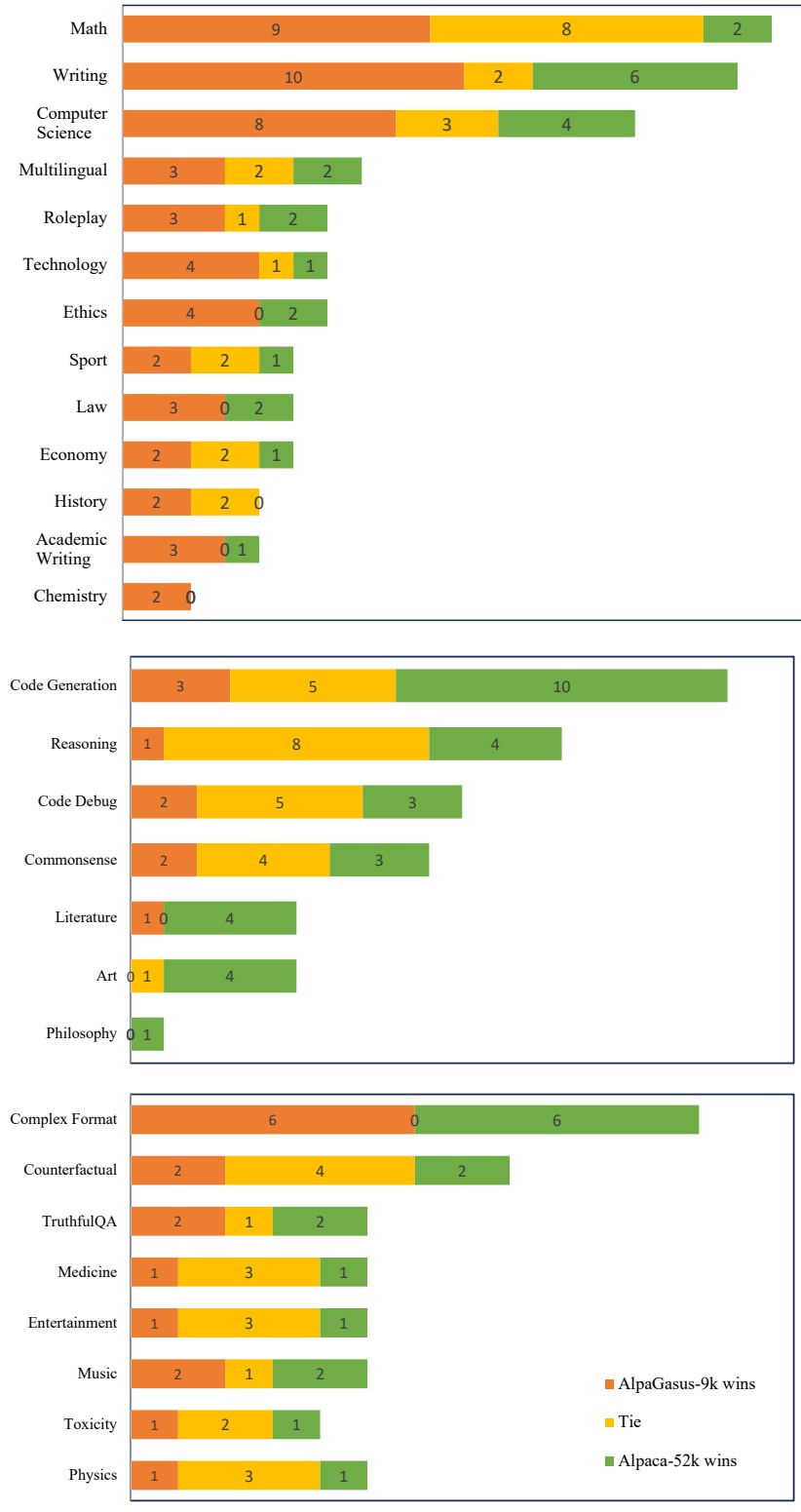

Figure 25: Fine-grained evaluation of ALPAGASUS-9k(7B) vs. ALPACA-52k(7B) on categories of the WizardLM test set.

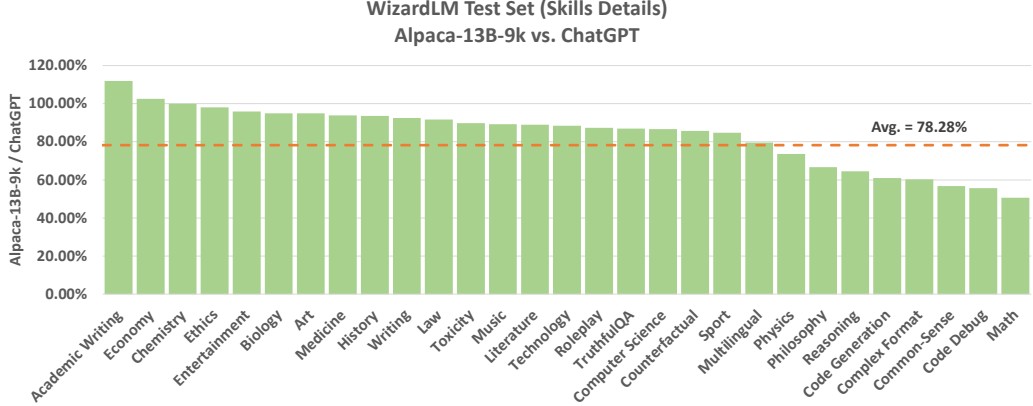

Figure 26: Compare with ChatGPT. Achieve average 78.26% capacity of ChatGPT on all 29 skills.

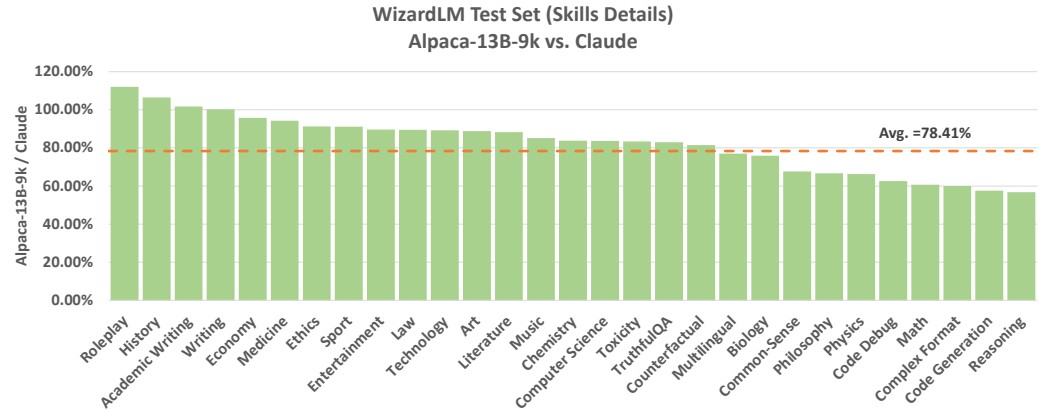

Figure 27: Compare with Claude-v1. Achieve average 78.41% capacity of ChatGPT on all 29 skills.

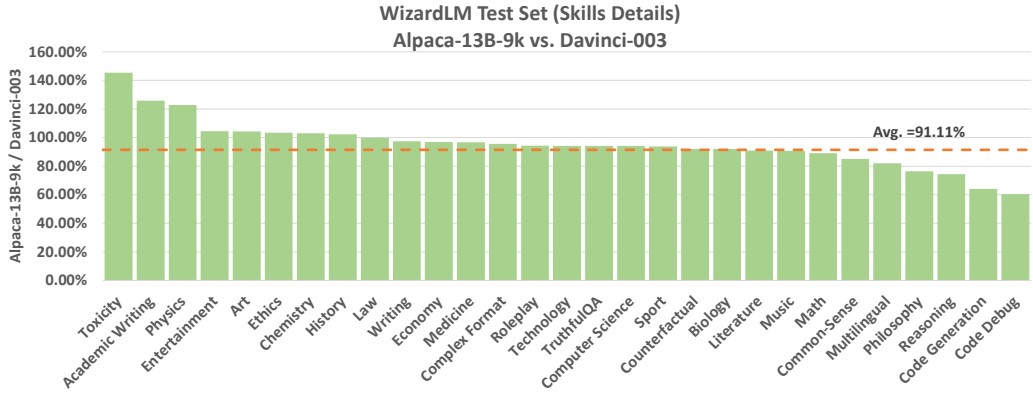

Figure 28: Compare with Davinci-003. Achieve an average 91.11% capacity of ChatGPT on all 29 skills.

## J   HUMAN STUDY

We conduct the human study among three different users. The evaluation interface is shown as Table 15:

---

You'll be presented with a series of questions. For each question, two answers will be provided. Your task is to read both answers carefully and decide which one you believe is better. When judging, consider:
**Relevance:** Does the answer directly address the question?
**Completeness:** Is the answer comprehensive?
**Coherence:** Is the answer logically structured and easy to understand?
**Accuracy:** Is the information provided in the answer correct?

**Question:**
<QUESTION>

**Answer A:**              **Answer B:**
<ANSWER A>              <ANSWER B>

**Comparing these two answers, which answer is better?**
1. Answer A is significantly better.
2. Answer B is significantly better.
3. Neither is significantly better.

---

Table 15: Human annotation interface.

We show more detailed results of human evaluations in Fig. 29.

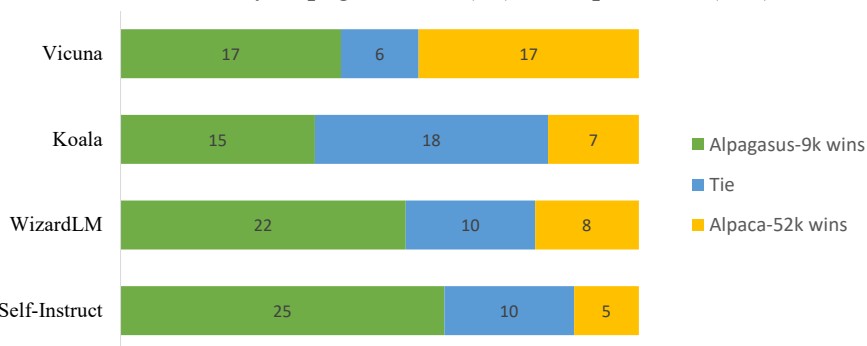

Figure 29: The detailed results of human study.

## K   LIMITATIONS

**Model Size.** In our experiments, we evaluated our IFT strategy by training models of two different sizes, 7B and 13B, since they are the most common sizes for recent open-source LLMs. We plan to extend this study to larger model sizes such as 33B, 65B, or even 175B, and verify whether the same conclusion still holds, i.e., a small subset of high-quality data selected by our method can improve the instruction-finetuned model. We leave analysis on the IFT of larger models as future work.

