# OpenReview forum: "AlpaGasus: Training a Better Alpaca with Fewer Data"
_ICLR.cc/2024/Conference — ICLR 2024 poster_

### Official Review · Reviewer_egVm · 2023-10-25

**Soundness:** 2 fair
**Presentation:** 3 good
**Contribution:** 1 poor
**Rating:** 6
**Confidence:** 5

**Summary:**

This paper presents a data filtering method based on ChatGPT to filter out the low-quality instruction-finetuning data. With the proposed mechanism, the authors demonstrate using only 9k high-quality data from ALPACA can achieve good performance. The proposed method is evaluated by ChatGPT and human on several language models.

**Strengths:**

* The paper is generally well-written and clear.
* This paper again demonstrates that the data quality in instruction tuning is more important than quantity.
* Many ablation experiments are provided in Appendix.

**Weaknesses:**

* As the authors stated in the limitation part, only 7B and 13B models are verified. I understand that this is mainly due to the computational constraint, but one concern is that similar observations may not scale on larger models.
* Knowing that the LLM models is sensitive to the prompt,  more prompt templates on data filtering are expected to be studied.
* The evaluation is mostly based on ChatGPT as the judge and the "Win-Tie-Lose" metric. More evaluation results are expected, especially on benchmarks related to adversarial and out-of-domain robustness. Evaluating solely on the curated instruction set is not enough to demonstrate the effectiveness of the data filtering method on other dimensions.

**Questions:**

* Regarding the second point of weakness. For example, does the granularity of the score affect the performance? What's the effect of system prompts on final performance?
* Regarding the third point of weakness. Does the reduced quantity of data have a detrimental effect on the robustness of tuned models?

---

> ### Author Response · Authors · 2023-11-18
>
> **Q1:** Does the granularity of the score affect the performance?
>
> **A1:** The granularity of ChatGPT is 0.5 and the granularity of Claude2 is 1. The score distributions are shown in Figure 5 and 14, respectively. It should be noted this granularity difference— 1 vs 0.5 —is inherent to the models' outputs in response to rating prompts. Our analysis reveals that both granularity levels effectively contribute to the filtration and refinement of data.
>
> **Q2:** What's the effect of system prompts on final performance?
>
> **A2:** Unlike ChatGPT, Claude2 operates without utilizing system prompts. Despite this difference, Claude2 demonstrates considerable proficiency in executing filtering tasks (Figure 14). This suggests that the absence of system prompts does not significantly impede its performance efficiency.
>
> **Q3:** Can the method scale up to larger models?
>
> **A3:** Due to the limited resouces we have, we run the experiments based on the LoRA and show that our filtering method can still work when the model size comes to 30B. We train the models on 8xRTX A6000 for ~18 hours for Alpaca and ~3.5 hours for AlpaGasus, with 128 batch size, 1e-5 learning rate, 1024 max length, and 5 epochs.
>
> Here is the instruction-following evaluations.
> Alpagasus-30B(9k data) vs. Alpaca-30B(52k data)
> | Testset | Win | Tie | Lose |
> |---     | ---    | ---     | ---    |
> | Vicuna | 20 | 50 | 10 |
> | Koala | 38 | 113 | 29 |
> | WizardLM | 41 | 141 | 36 |
> |Self-Instruct | 43 | 172  | 37 |
>
>
> Alpagasus-30B(9k data) vs. Alpaca-30B(9k random data)
> | Testset | Win | Tie | Lose |
> |---     | ---    | ---     | ---    |
> | Vicuna | 18 | 53 | 9 |
> | Koala | 35 | 114 | 31 |
> | WizardLM | 47 | 148 | 23 |
> |Self-Instruct | 50 | 165  | 37 |
>
> From the table, we can see that when the model size goes up to 30B, our Alpagasus can still perform better than the Alpaca-52k with only 9k filtered data.

---

> ### Author Response · Authors · 2023-11-18
>
> **Q4:** Different prompts are expected to be studied.
>
> **A4:**
> In fact, we did not do any kind of prompt engineering. We also consider that different dimensions could have different criteria in the rating, thus besides the accuracy-focused prompt used for the main experiments, we also tried a helpfulness-focused prompt as well and the results are shown in Appendix A.5. The follow-up work Humpback[1] uses different prompts that share the same spirit for filtering the IFT data and they achieve good results as well. These all prove that the good performance of our method doesn't depend on specific prompts. We will add more analysis on different prompt variations in our final version.
>
> **Q5:** What is the agreement between the scores obtained using Claude2 as the LLM filter and those using ChatGPT?
>
> **A5:** ChatGPT filters 9,229 examples from original 52k Alpaca data and Claude2 filters 8,080 examples. We found these two sets have 5,664 shared examples and the agreement between ChatGPT and Claude2 is 70.1%.
>
> **Q6:** Are the results in Figure 14 also evaluated by GPT-4?
>
> **A6:** Yes, the evaluation in Figure14 is conducted by GPT-4. Claude-2 is another LLM filter and we observe consistent gain which proves our method is versatile across LLM filters (it works well on both ChatGPT and Claude2.)
>
> [1] Self-alignment with Instruction Backtranslation. https://arxiv.org/abs/2308.06259

---

> ### Author Response · Authors · 2023-11-18
>
> **Q7:** Does the reduced quantity of data have a detrimental effect on the robustness of tuned models?
>
> **A7:** Thanks for your question. We find that the robustness is not affected. Following [2], which designs a benchmark to evaluate how well LLMs follow developer-provided rules in the face of adversarial inputs, we test the adversarial robustness of the AlpaGasus-7B(9k), Alpaca-7B(52k), and Alpaca-7B(9k random). The results are shown as follows:
> |               | Avg.      | Indirect   | Just-Ask      | Legalese      | Obfs   | Rule-C| Sim    |
> |---------------|---------------|---|---------------|---------------|---------------|---------------|---------------|
> | Alpagasus    | **38.12**   | 37.97   | 36.36   | 35.23   | 38.96   | 38.64   | 41.67   |
> | Alpaca | 38.02 | 38.23 | 40.68 |	32.95 |	38.96 |	38.63 |	41.67 |
> |Alpaca(random) | 35.58 |	34.22 |	36.36 |	36.36 |	38.96 |	35.61 |	35.71 |
>
> where `Indirect`, `Just-ask`, `legalese`, `Obfuscation`, `Rule Change`, and `Simulation` are six different adversarial strategies trying to break the rules.
>
> As the table shows, Alpagasus performs the best among the three models, which proves its effectiveness on adversarial robustness.
>
> Here, we also provide the detailed category results for each model.
> | Test Case  | AlpaGasus      | Alpaca(random)     | Alpaca     |
> |-----|---|---|---|
> | Encryption                        | 89.744        | 82.051        | 92.308     |
> | Integrity                         | 0             | 0             | 0             |
> | AccessControl                     | 100           | 92.683        | 92.683        |
> | Commitment_reject                 | 0             | 0             | 0             |
> | Commitment_verify                 | 0             | 0             | 0             |
> | Confidentiality                   | 97.561        | 92.683        | 95.122        |
> | Authentication_provide            | 0             | 0             | 0             |
> | Authentication_withholdpassword   | 100           | 94.872        | 94.872        |
> | Authentication_withholdsecret     | 87.179        | 89.744        | 92.308        |
> | Millionaires_identify             | 0             | 0             | 0             |
> | Millionaires_withhold             | 97.436        | 84.615        | 94.872        |
> | DiningCryptographers              | 0             | 0             | 0             |
> | Hodor                             | 0             | 0             | 0             |
> | ForbiddenWord                     | 66.667        | 46.154        | 79.487        |
> | RockPaperScissors                 | 0             | 0             | 0             |
> | AnimalSounds                      | 0             | 0             | 0             |
> | Questions_confirm                 | 0             | 0             | 0             |
> | Questions_withhold                | 100           | 100           | 97.436        |
> | BinarySearch_confirm              | 0             | 0             | 0             |
> | BinarySearch_withhold             | 100           | 100           | 97.436        |
> | SimonSays_execute                 | 0             | 0             | 0             |
> | SimonSays_reject                  | 0       | 0          | 0             |
> | **Average**                       | **38.12** | **35.58** | **38.02** |
>
>
> Additionally, the follow-up work[3] confirms that our filtering method could filter the backdoor examples in the training data, thereby confirming our approach to enhancing the safety of the IFT data by filtering the biased responses.
>
> In conclusion, AlpaGasus serves a dual purpose: it not only enhances the instruction-following capabilities of models but also plays a significant role in advancing their safety alignment.
>
> [2] Can LLMs Follow Simple Rules? https://arxiv.org/abs/2311.04235
>
> [3] Backdooring Instruction-Tuned Large Language Models with Virtual Prompt Injection. https://arxiv.org/abs/2307.16888

---

> ### Author Response · Authors · 2023-11-21
>
> Dear Reviewer,
>
> We haven't heard from you since sending you our rebuttal. Since we are approaching the last day of the reviewer-author discussion, it would be really nice of you to confirm if your initial concerns (most of them are clarification questions) were successfully addressed by our rebuttal. We are looking forward to your feedback and we kindly expect that you can raise the score if all your main concerns have been resolved.
>
> Thanks!
>
> Best regards,
>
> Authors

---

> > ### Comment · Reviewer_egVm · 2023-11-22
> >
> > Thanks for the authors' detailed response. Most of my concerns have been addressed. I would like to increase my score. However, in my humble opinion, this work may present limited novelty and significance given the similar works on data curation of tuning LLMs and common filtering approaches with ChatGPT.

---

> ### Author Response · Authors · 2023-11-22
> **Thank you! Some clarification of the novelty**
>
> Dear Reviewer #egVm,
>
> It is great to hear that most of your concerns have been addressed! Thank you so much for raising your score! Your constructive input remains invaluable to us, and we appreciate your dedication to enhancing the quality of our manuscript.
>
> **Regarding the novelty and significance**, we would like to emphasize: to the best of our knowledge, we are the **first to automatically filter IFT data and improve the IFT of the LLMs** by prompting LLMs as the filters, e.g., ChatGPT and Claude2. This method is novel in selecting a high-quality subset of IFT data, which not only improves the LLMs' performance but also presents significant computational advantages.
>
> **Many follow-up works.** We acknowledge that since we released our paper on Arxiv, many follow-up studies [1,2,3,4] have emerged, some of which have utilized our method as a baseline or integrated it into their algorithms. While these subsequent works may give an impression that our work is less novel, it is important to emphasize that these studies build upon our foundational method. Our work served as a pioneering effort in this area, setting a precedent for further exploration and innovation.
>
> We appreciate your attention to this matter and hope this clarification underscores the originality and significance of our contribution to the field.
>
> [1] What Makes Good Data for Alignment? A Comprehensive Study of Automatic Data Selection in Instruction Tuning. https://openreview.net/forum?id=BTKAeLqLMw
>
> [2] From Quantity to Quality: Boosting LLM Performance with Self-Guided Data Selection for Instruction Tuning. https://arxiv.org/abs/2308.12032
>
> [3] InstructionGPT-4: A 200-Instruction Paradigm for Fine-Tuning MiniGPT-4. https://arxiv.org/pdf/2308.12067.pdf
>
> [4] Self-Alignment with Instruction Backtranslation. https://arxiv.org/abs/2308.06259

---

### Official Review · Reviewer_BizM · 2023-10-30

**Soundness:** 2 fair
**Presentation:** 2 fair
**Contribution:** 2 fair
**Rating:** 6
**Confidence:** 4

**Summary:**

This paper proposes a data selection method to remove noisy or low-quality data from the 52k Alpaca data for instruction-finetuning (IFT). Specifically, the authors use ChatGPT to evaluate the quality of the Alpaca 52K instruction tuning dataset and filter out the data samples of low quality. The experimental results show that LLaMA finetuned on the selected dataset outperforms the LLaMA finetuned on the whole Alpaca 52K dataset, as evaluated by GPT-4.

**Strengths:**

- The paper is generally clear to read and easy to follow.
- The demonstrated method for data selection is simple and easy to understand.
- The experimental results show some improvement over the baseline finetuned on the whole dataset.
- The overall experiments are quite extensive.

**Weaknesses:**

- The authors claim that it costs less to finetune with the filtered dataset by comparing the computation costs in the finetuning stage. However, the authors didn't consider the costs in the data filtering stage where they used the ChatGPT to get the scores for the data samples, which also cost money, computation, and time.
- Alpagasus is partially weaker than Alpaca (as in Table 2), especially on the large MMLU dataset. The datasets where Alpagasus outperforms Alpaca seem to be small datasets, mostly consisting of a few hundred samples. Moreover, the improvement of Alpagasus over Alpaca is marginal. These issues make the method of Alpagasus not quite convincing.
- Since the data quality scores are evaluated by ChatGPT (using the OpenAI GPT-4 API?) and the final model output evaluation on test sets is performed by GPT-4, it may lead to a bias towards the GPT-4's preference in the finetuning and testing stages. In other words, GPT-4 selects the data samples it prefers, which can naturally be used to finetuned a model with the same output characteristics that is preferred by GPT-4.
- It is unclear how many runs are performed for the random splits. The standard deviations of the results are also needed in the paper.
- In some test sets, the results of the model finetuned on a random split are comparable to, sometimes even better than, those finetuned on the whole dataset. This observation may suggest that the instruction tuning actually suffers from overfitting to the whole training set so the results obtained by training on random splits are even better.

**Questions:**

- What are the standard deviations of the results obtained from the random data splits? And how many runs?
- Are the results in Figure 14 also evaluated by GPT-4?
- What is the agreement between the scores obtained using Claude2 as the LLM filter and those using ChatGPT?
- How reliable is the LLM filter?

---

> ### Author Response · Authors · 2023-11-18
> **Data filtering is a one-off expense and the filtered data can be applied to different base models**
>
> **Q1:** The authors claim that it costs less to finetune with the filtered dataset by comparing the computation costs in the finetuning stage. However, the authors didn't consider the costs in the data filtering stage where they used the ChatGPT to get the scores for the data samples, which also cost money, computation, and time.
>
> **A1:** We acknowledge the costs associated with using ChatGPT for data filtering. Nonetheless, the overall expense of utilizing the ChatGPT API for filtering 52,000 data samples is approximately **50 dollars, requiring around 1.5 hours**. Our approach is both more efficient and more economical than human annotation, which typically costs around 25 dollars per hour[1] and takes at least **433 hours and 10,825 dollars** to filter the same amount of data. Assume that one person takes 30 seconds to rate one data sample, the total time will be:
>
> `T=(52,002 x 30s)  / (60 x 60)=433.35 hrs`
>
>
> Crucially, the investment in IFT (instruction finetuning) data filtering is a one-off expense. In practice, base models are subject to continuous evolution, as exemplified by Meta's release of LLaMA-1[1] in February and LLaMA-2[2] in July. Our research demonstrates that our filtered data effectively supports both LLaMA-1 (see Figure 5) and LLaMA-2 (see Figure 15), indicating its applicability across various model iterations.
>
> Furthermore, training costs escalate with model size (7B, 13B models), as detailed in Section 7 of our paper. Our methodology can **reduce the training costs for 13B models from 225.28 dollars to 40.96 dollars**[3]. The application of our method to larger models offers even greater cost savings, as illustrated in the accompanying table, which provides an estimated cost analysis. Moreover, Applying our method to ultra-large models like BLOOM-176B could have further potential savings on instruction-tuning costs.
>
> | Model Size | Data Size | # GPUs | Epoch | Time | Cost    |
> |------------|-----------|--------|-------|------|---------|
> | 7B         | 9k        | 4      | 3     | 14m  | $ 4.78 |
> | 7B         | 52k       | 4      | 3     | 80m  | $ 27.31|
> | 13B        | 9k        | 8      | 5     | 1h   | $ 40.96 |
> | 13B        | 52k       | 8      | 5     | 5.5h | $ 225.28|
> | 30B         | 9k        | 16      | 10  | 2.18h  | $ 178 |
> | 30B         | 52k       | 16      | 10  | 12h  | $ 901 |
>
> [1] https://www.ziprecruiter.com/Salaries/Data-Annotation-Salary
>
> [2] LLaMA-2: Open Foundation and Fine-Tuned Chat Models: https://arxiv.org/abs/2307.09288
>
> [3] LIMA. https://arxiv.org/abs/2305.11206

---

> ### Author Response · Authors · 2023-11-18
>
> **Q2:** Alpagasus is partially weaker than Alpaca (as in Table 2), especially on the large MMLU dataset. The datasets where Alpagasus outperforms Alpaca seem to be small datasets, mostly consisting of a few hundred samples. Moreover, the improvement of Alpagasus over Alpaca is marginal. These issues make the method of Alpagasus not quite convincing.
>
> **A2:**
> + “Partially weaker in MMLU”: MMLU contains hundreds of diverse tasks NOT designed to specifically evaluate instruction-following capability, which is the goal of instruction-finetuning (IFT) and the focus of training data we filtered in this paper. Moreover, MMLU is a knowledge-intensive benchmark, but we do not expect to learn new knowledge during IFT. Instead, we expect IFT to refine the models' ability to comprehend and respond to complex user queries, as established in LIMA[3]. Our goal with AlpaGasus is to maintain, rather than significantly enhance performance in the contexts of MMLU. Notably, AlpaGasus outperforms all models trained with randomly selected samples, across all model scales, and independent of the IFT dataset used. This is a significant achievement, underscoring the efficacy of our approach.
> + “Improvement marginal in the benchmark”: As we emphasize in the general response to all reviewers, instruction-following is a more important evaluation for instruction tuning and AlpaGasus consistently demonstrates gains in instruction-following scores.  Furthermore, the follow-up work[4] shows our method contributes to enhancing model safety.
> + When the model sizes go up, high-quality data becomes more important. So our data filtering brings more improvement over random selection. For example, on Dolly, the 13B(3k) model is better than 13B(15k) on all four benchmark datasets.
>
> [3] LIMA. https://arxiv.org/abs/2305.11206
>
> [4] VPI: https://arxiv.org/abs/2307.16888

---

> ### Author Response · Authors · 2023-11-18
>
> **Q3:** Since the data quality scores are evaluated by ChatGPT (using the OpenAI GPT-4 API?) and the final model output evaluation on test sets is performed by GPT-4, it may lead to a bias towards the GPT-4's preference in the finetuning and testing stages. In other words, GPT-4 selects the data samples it prefers, which can naturally be used to finetune a model with the same output characteristics that are preferred by GPT-4.
>
> **A3:**
>
> + **Clarification:** It is important to note that in our method, we utilized GPT-3.5-turbo for data filtering and GPT-4 for automatic evaluation purposes. These are distinct models with different functionalities in our framework. Besides, the use of GPT-4 for automatic evaluation is not unique to our study. Similar model-based evaluations have been previously employed [5].
> + **Human study:** To further validate our findings, we conducted human-based evaluations as detailed in Section 4. It shows AlpaGasus' answers are preferred when compared with the answers generated by Alpaca.
> + **Other LLM filter** To address concerns regarding any bias of GPT-3.5-turbo, we also apply Claude2 as our LLM filter (Appendix A.2), and we use GPT-4 as the judge. Our follow-up work Humpback[6] shows the LLaMA-2-70B model could self-filter its responses by applying a similar method. To conclude, We demonstrate the effectiveness of our method on different models and it can help mitigate potential bias concerns.
>
> [5] Judging LLM-as-a-Judge with MT-Bench and Chatbot Arena. https://arxiv.org/abs/2306.05685

---

> ### Author Response · Authors · 2023-11-18
>
> **Q4:** It is unclear how many runs are performed for the random splits. The standard deviations of the results are also needed in the paper.
>
> **A4:** Thanks for your question! We generate another random split with a different random seed and evaluate the model trained with it. The benchmark results are shown in the following table.
> | Dataset | BBH |  DROP | Humaneval | MMLU |
> | ---    | --- | ---    | ---     | ---    |
> | Alpaca(9k-random)-v1 | 31.89 | 25.88 | 11.59 | 36.93 |
> | Alpaca(9k-random)-v2 | 31.71 | 25.63 | 11.72 | 36.71 |
> | AlpaGasus        | 33.76 | 26.03 | 12.20 | 38.78 |
>
> We can conclude that our AlpaGasus can significantly perform better than the models instruction-tuned with randomly selected data.
>
>
> We also show the instruction-following evaluations:
>
> |Win score| Vicuna | Koala | WizardLM | Self-Instruct |
> |---     | ---    | ---     | ---    | ---        |
> |Ours-9k vs. 9k-random-v1 | 1.213 | 1.022 | 1.060 | 1.056 |
> |Ours-9k vs. 9k-random-v2 | 1.188 | 1.033 | 1.069 | 1.071 |
>
>
> The winning score is calculated by:  `1 + (#Win−#Lose) / (#Testset)`,
>
> which is also shown in the caption of Figure 1.

---

> ### Author Response · Authors · 2023-11-21
>
> Dear Reviewer,
>
> We haven't heard from you since sending you our rebuttal. Since we are approaching the last day of the reviewer-author discussion, it would be really nice of you to confirm if your initial concerns (most of them are clarification questions) were successfully addressed by our rebuttal. We are looking forward to your feedback and we kindly expect that you can raise the score if all your main concerns have been resolved.
>
> Thanks!
>
> Best regards,
>
> Authors

---

### Official Review · Reviewer_qmSJ · 2023-10-31

**Soundness:** 3 good
**Presentation:** 3 good
**Contribution:** 2 fair
**Rating:** 6
**Confidence:** 4

**Summary:**

This paper proposes filtering low-quality instruction data by an LLM judger (GPT-4 specifically). The authors show that fine-tuning an LLM on the filtered data offers performance better than on the entire raw dataset.

**Strengths:**

The idea of filtering low-quality data is reasonable. The experiments are extensive: The authors compare the the model fine-tuned on the filtered datasets on several types of evaluation suites.

**Weaknesses:**

- The filtering model ablation study: The performance of Alpagasus seems to highly depend on the quality of the filtering model. How often does GPT-4 as the filtering model makes a mistake? What is the effect of using different filtering model (GPT-3.5, Claude-2, Llama, etc)? As of now, the paper does not offer any study or motivation of using GPT-4 as the filtering model, and simply assumes that is the best choice.

**Questions:**

See above.

---

> ### Author Response · Authors · 2023-11-18
> **Motivation**
>
> Thanks for your feedback! Firstly, we would like to reemphasize our motivation for filtering: the low-quality data in IFT could be harmful to the model (we show some bad examples in Alpaca dataset in Figure 2). The previous method, e.g., Alpaca-cleaned, relies on human labor to check the quality of the data which is time-consuming and expensive. Thus, we propose an automatic, efficient, and scalable filtering method that can scale up to large-size datasets.

---

> ### Author Response · Authors · 2023-11-18
>
> **Q1:** The performance is highly dependent on the filtering models.
>
> **A1:** In our paper, we use two different filtering models, i.e., ChatGPT and Claude2 (Appendix A.5). Comprehenvise instruction-following evaluations (GPT4-based and human study) across four different test sets show the effectiveness of our filtering method. As for using the open-sourced LLMs for filtering, our follow-up work, Humpback[1], even proves that the 70B-LLaMA model can self-filter by using prompts with a proper threshold. These results all show evidence that our method is not highly dependent on the models.
>
>
> **Q2** The paper does not offer any study or motivation for using GPT-4 as the filtering model and simply assumes that GPT-4 is the best choice.
>
> **A2:** Thanks for your question! We would like to point out that the claim "using GPT-4 as the filtering model" is **not accurate**. In fact, we do not use GPT-4 as our LLM filter: in the captions of Figure 2, 3, and Section 2.1, we clearly mention that the LLM filter we used in the main paper is ChatGPT(GPT-3.5-turbo). Furthermore, we **never** assume that "GPT-4 is the best choice". In Appendix A.2., we show our method can also work well even if we choose Claude2 as our LLM filter. Our follow-up work, Humpback[1], proves that the model can self-filter by using prompts and setting up a proper threshold, where they use the LLaMA-70B model. To conclude, we are the **first to propose an automatic and scalable filtering method for IFT data filtering** and it can work well across different filtering models (ChatGPT, Claude, and LLaMA).
>
> **Q3:** How often does GPT-4 as the filtering model make a mistake?
>
> **A3:** We show high-quality examples in Table 7,8,11,12 and low-quality examples in Table 10,14. We would like to highlight that they are not cherry-picked examples and we just randomly select some pairs from each level of scores. We also examine 400 examples manually, i.e., 200 examples from the high-quality 9k data and another 200 examples from the low-quality 43k, but we haven't found any mistakes the LLM filter makes.
>
> The comprehensive instruction-following evaluations shown in the paper also prove that our filtering method is reliable on both human-written/machine-generated datasets.
>
>
> [1] Self-alignment with Instruction Backtranslation. https://arxiv.org/abs/2308.06259

---

> ### Author Response · Authors · 2023-11-22
>
> Dear Reviewer,
>
> We haven't heard from you since sending you our rebuttal. Since we are approaching the last day of the reviewer-author discussion, it would be really nice of you to confirm if your initial concerns (most of them are clarification questions) were successfully addressed by our rebuttal. We are looking forward to your feedback and we kindly expect that you can raise the score if all your main concerns have been resolved.
>
> Thanks!
>
> Best regards,
>
> Authors

---

### Author Response · Authors · 2023-11-18
**General Response**

Dear reviewers,

Thanks for all your constructive comments! We hope our responses address your concerns. Please feel free to raise further concerns and we are open to discuss.

**Motivation for Data Filtering** First of all, we would like to clarify our motivation for doing this instruction-tuning(IFT) data filtering work. As we mentioned in the Introduction: “Alpaca-Clean[1] is the pioneer of filtering bad data in Alpaca dataset but it requires humans fully involved in examining and filtering bad data”. This method cannot scale up to 1 million or 1 billion data samples since human annotations are expensive and time-consuming. Thus, we would like to have a scalable, convenient way, without human efforts, to filter the IFT data.

**Comparison with FLAN-PALM[2]:** Contrary to our method, FLAN-PALM[2] collected the data from 1.8k tasks and performed multi-task instruction tuning. Their findings suggest that increasing the amount of data enhances performance. In contrast, our research demonstrates the feasibility of selecting a high-quality data subset that surpasses the performance of models trained on the entire dataset. It should be noted that this observation does not contradict FLAN-PALM's findings but rather complements them by emphasizing that the enhancement in model performance is largely attributable to high-quality data. Furthermore, our research underscores the detrimental effects of incorporating low-quality data, thereby highlighting the critical role of data filtering in IFT.

**IFT does not inject new knowledge:** Secondly, we would like to clarify that the instruction-finetuning~(IFT) does not inject new knowledge into the LLMs since compared with the huge pretraining corpus, e.g., LLaMA-2[3] trained on 2000B tokens of data, the IFT data, which is usually <0.2B tokens (calculated using Alpaca-52k data), are usually not intended to inject new knowledge. LIMA[4] also shares the same understanding of IFT with us: IFT focuses on learning the complex format of the questions asked by humans. Thus, we believe instruction-following evaluation is more important and IFT should hardly hurt the performance of LLMs on knowledge-intensive benchmarks.

**Importance of Data Quality** Thirdly, our results show that data quality is more important than quantity and our method proves effective across different rating dimensions (e.g., accuracy and helpfulness), LLM filters (e.g., ChatGPT and Claude-2), base model families (e.g., LLaMA-1 and LLaMA-2), model sizes (e.g., 7B and 13B), dataset types (e.g., machine-generated and human-written) under different evaluations (e.g., GPT-4-based evaluations and human study).

**The evaluation LLM is different from LLM filters:** In our experiment, we use ChatGPT and Claude2 as our LLM filters. For evaluations, we use GPT-4 for instruction-following evaluations.

**Evaluations on Adversarial Robustness:**  We conduct extra experiments on llm-rule benchmark[5], which aims to evaluate how well LLMs follow developer-provided rules in the face of adversarial inputs. We test the adversarial robustness of the AlpaGasus-7B(9k), Alpaca-7B(52k), and Alpaca-7B(9k random). The results are shown as follows:
|               | Avg.      | Indirect   | Just-Ask      | Legalese      | Obfs   | Rule-C| Sim    |
|---------------|---------------|---|---------------|---------------|---------------|---------------|---------------|
| Alpagasus    | **38.12**   | 37.97   | 36.36   | 35.23   | 38.96   | 38.64   | 41.67   |
| Alpaca | 38.02 | 38.23 | 40.68 |	32.95 |	38.96 |	38.63 |	41.67 |
|Alpaca(random) | 35.58 |	34.22 |	36.36 |	36.36 |	38.96 |	35.61 |	35.71 |

where `Indirect`, `Just-ask`, `legalese`, `Obfuscation`, `Rule Change`, and `Simulation` are six different adversarial strategies trying to break the rules.

As the table shows, Alpagasus performs the best among the three models, which proves its effectiveness on adversarial robustness.

[1] Alpaca-clean: https://github.com/gururise/AlpacaDataCleaned/

[2] FLAN-PALM https://arxiv.org/pdf/2210.11416.pdf

[3] LLaMA2 technical report. https://arxiv.org/pdf/2307.09288.pdf

[4] Less is more for alignment. https://arxiv.org/pdf/2305.11206.pdf

[5] Can LLMs Follow Simple Rules? https://arxiv.org/abs/2311.04235

---

### Meta-Review · Area_Chair_Xmze · 2023-12-12

**Metareview:**

This paper proposes a data filtering method to remove noisy or low-quality data from the 52k Alpaca data for instruction-finetuning. After rebuttal, it received scores of 666.

On the one hand, reviewers commented that the paper is well written, the method is simple and easy to understand, and experiments are extensive. On the other hand, the AC also agrees that this work presents limited novelty given the similar works on data curation of tuning LLMs and common filtering approaches with ChatGPT.

Overall, all the reviewers are generally positive about the paper, and the AC would like to recommend acceptance by the end.

**Justification For Why Not Higher Score:**

The paper presents limited novelty, as the idea of filtering low-quality data to improve instruction finetuning is indeed simple, but may not be novel enough.

**Justification For Why Not Lower Score:**

All the reviewers are happy about the paper, and though the idea is simple, experiments are comprehensive to show this method works.

---

### Decision · Program_Chairs · 2024-01-16

Accept (poster)